# Towards Scalable and Stable Parallelization of Nonlinear RNNs

**Xavier Gonzalez[1,2], Andrew Warrington[1,2,3], Jimmy T.H. Smith[2,4,5], Scott W. Linderman[1, 2]**
[1]Department of Statistics, Stanford University.
[2]Wu Tsai Neurosciences Institute, Stanford University.
[3]GE Healthcare.   [4]ICME, Stanford University.   [5]Liquid AI.
{xavier18,scott.linderman}@stanford.edu

## Abstract

Transformers and linear state space models can be evaluated in parallel on modern hardware, but evaluating nonlinear RNNs appears to be an inherently sequential problem. Recently, however, Lim et al. [1] developed an approach called DEER, which evaluates nonlinear RNNs in parallel by posing the states as the solution to a fixed-point problem. They derived a parallel form of Newton's method to solve the fixed-point problem and achieved significant speedups over sequential evaluation. However, the computational complexity of DEER is cubic in the state size, and the algorithm can suffer from numerical instability. We address these limitations with two novel contributions. To reduce the computational complexity, we apply quasi-Newton approximations and show they converge comparably to Newton, use less memory, and are faster. To stabilize DEER, we leverage a connection between the Levenberg-Marquardt algorithm and Kalman smoothing, which we call ELK. This connection allows us to stabilize Newton's method while using efficient parallelized Kalman smoothing algorithms to retain performance. Through several experiments, we show that these innovations allow for parallel evaluation of nonlinear RNNs at larger scales and with greater stability.

## 1   Introduction

Parallel computation has helped fuel the rise of deep learning [2]. Architectures such as transformers [3] and linear RNNs [4–8] are specifically designed to allow parallelization over the length of the input sequence. However, most conventional nonlinear RNNs (e.g. Elman RNNs, GRUs [9], LSTMs [10] etc.) are not readily parallelizable over the sequence length due to their sequential architecture. Thus, they do not benefit as much from parallel hardware. Nonetheless, these nonlinear RNN architectures are still used widely across the scientific community [11–13]. Furthermore, recent work has suggested that linear RNNs (and transformers) are fundamentally limited in their expressivity compared to nonlinear RNNs [14]. Finally, nonlinear RNNs continue to be of significant interest in computational and theoretical neuroscience as models of neural systems [15–22]. Therefore, scalable and stable parallelization methods for nonlinear RNNs offer significant benefits across many fields.

Towards this goal, Lim et al. [1] proposed DEER, a method for evaluating a nonlinear RNN in parallel. DEER casts inference as finding the solution of a fixed-point equation designed specifically to capture the nonlinear dynamics of the RNN. Newton's method is used to solve the resulting fixed-point equation. With good initialization, Newton's method enjoys quadratic convergence rates [23, Chapter 11]. Lim et al. [1] also show that the inversion of the structured Jacobian matrix required by Newton's method can be cast as an associative parallel scan [24]. DEER therefore reduces the evaluation runtime over sequential evaluation by as much as factor of twenty.

38th Conference on Neural Information Processing Systems (NeurIPS 2024).

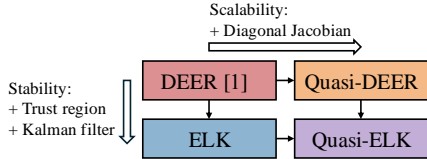

**Figure 1:** Overview of the paralleliz-able methods we consider in this paper. We introduce diagonal approximations to improve complexity (quasi-DEER, Section 4.1) and link to Kalman filtering and trust regions to improve stability (ELK, Section 4.2). We combine these ideas in quasi-ELK (Section 4.2).

**Table 1:** Description of the relative strengths and weaknesses of the five evaluation methods we consider. We discuss further in Section 7.

| Method | Desiderata | | | |
|---|---|---|---|---|
| | Parallel | Work | Memory | Stability |
| Sequential | No | $\mathcal{O}(TD^2)$ | $\mathcal{O}(D)$ | Very high |
| DEER [1] | Yes | $\mathcal{O}(TD^3)$ | $\mathcal{O}(TD^2)$ | Low |
| Quasi-DEER | Yes | $\mathcal{O}(TD)$ | $\mathcal{O}(TD)$ | Low |
| ELK | Yes | $\mathcal{O}(TD^3)$ | $\mathcal{O}(TD^2)$ | High |
| Quasi-ELK | Yes | $\mathcal{O}(TD)$ | $\mathcal{O}(TD)$ | Moderate |

However, DEER also inherits the weaknesses of Newton's method and parallel scans. The first weakness is *scalability*. Let $D$ denote the state dimension and $T$ denote sequence length. Using a parallel scan to evaluate updates from Newton's method, DEER inherits $\mathcal{O}(TD^2)$ memory complexity and $\mathcal{O}(TD^3)$ computational work [24]. These costs can be prohibitive in practical deep learning settings. The second limitation of DEER is *numerical stability*, inherited from Newton's method. In general, *undamped* Newton's method does not provide global convergence guarantees and in practice often diverges [23]. We seek to ameliorate both of these weaknesses.

To do this, we leverage two techniques: quasi-Newton approximations and trust regions. Quasi-Newton approximations are a common adaptation of Newton's method, where approximate, but faster and less memory intensive updates are used instead of exact Newton steps. Empirically, these methods often expedite convergence in terms of wall-clock time, even though more iterations are performed. We apply quasi-Newton approximations to reduce the memory and compute required by DEER, and we find accelerated convergence and reduced memory consumption. Secondly, we leverage a connection between Newton's method with a trust region and Kalman smoothing in sequential models [25]. This connection allows us to stabilize the Newton iteration by limiting the step size to the radius of the trust region, preventing large and numerically unstable steps. The update can be computed with a parallel Kalman smoother [26, 27], achieving a runtime that is logarithmic in the sequence length. We refer to DEER accelerated with a quasi-Newton approximation as quasi-DEER, and DEER stabilized with trust regions as "**E**valuating **L**evenberg-Marquardt via **K**alman" (ELK). We then combine these approaches to yield a fast and stable algorithm, which we call quasi-ELK.

Crucially, DEER, ELK, and their "quasi" variants are *algorithms* for parallelizing *any* discrete-time nonlinear dynamical system, including stateful architectures such as RNNs, that may or may not include stochasticity. We use "parallel" to refer to the fact that each iteration of our iterative algorithm operates on the *entire* $T$-length sequence (and not on each sequence element one at a time).

We outline the key contributions and organization of the paper here. First, we introduce background material, particularly focusing on DEER [1], in Sections 2 and 3. We then present three short novel proofs: that DEER is globally convergent; that this convergence is robust to modifications of the linearized dynamics (Proposition 1); and that there is a unique solution with no local minima (Appendices A.1 and A.2). We then introduce quasi-Newton approximations to DEER to improve efficiency (quasi-DEER, Section 4.1), and trust regions to stabilize DEER (ELK, Section 4.2) We also provide an interpretation of how trust regions stabilize the dynamics by damping the eigenvalues of the Jacobians (Appendix A.3). We show empirically that quasi-DEER remains accurate, with reduced runtime and memory consumption (Section 6). In regimes where DEER is numerically unstable or convergences slowly, we show ELK and quasi-ELK can enjoy fast, numerically stable convergence. We conclude by discussing the relative strengths and weaknesses of each method, providing guidance on how to select and tune them, and highlighting avenues for future research (Section 7). We provide our code at `https://github.com/lindermanlab/elk`.

## 2  Problem Statement

We consider nonlinear Markovian state space models, with the state at time $t$ denoted $\mathbf{s}_t \in \mathbb{R}^D$ and nonlinear transition dynamics $f : \mathbb{R}^D \to \mathbb{R}^D$. We denote the full sequence of $T$ states as $\mathbf{s}_{1:T} \in \mathbb{R}^{T \times D}$. Note that we primarily focus on the transition dynamics, so we suppress any (possibly random) input dependence of the model in the notation. However, the algorithms in this paper extend easily to these situations.

For any collection of candidate states $\{\mathbf{s}_t\}_{t=1}^T$ and an *initial state* $\mathbf{s}_0$ we can define the *residual*

$$\mathbf{r}(\mathbf{s}_{1:T}) := [\mathbf{s}_1 - f(\mathbf{s}_0),\ \mathbf{s}_2 - f(\mathbf{s}_1),\ \mathbf{s}_3 - f(\mathbf{s}_2), \dots, \mathbf{s}_T - f(\mathbf{s}_{T-1})] \in \mathbb{R}^{T \times D}. \tag{1}$$

This residual can be interpreted as the one-step error incurred by assuming the $t^{\text{th}}$ state is $\mathbf{s}_t$ instead of $f(\mathbf{s}_{t-1})$. The solution of the state space model, $\mathbf{s}_{1:T}^*$, is the only trace with zero residual. Equivalently, it is the unique solution to the fixed-point equation

$$\mathbf{r}(\mathbf{s}_{1:T}^*) = \mathbf{0}. \tag{2}$$

The conventional way to obtain $\mathbf{s}_{1:T}^*$ is to apply $f$ sequentially $T$ times. Sequential evaluation always yields a valid trace, but it requires $\mathcal{O}(T)$ sequential operations (i.e. computational depth or span), and hence does not fully leverage the capabilities of parallel hardware. We aim to compute $\mathbf{s}_{1:T}^*$ in sublinear time using parallel computation.

**Jacobian of the Residual**   For notational brevity, we overload $\mathbf{s}$ and $\mathbf{r}$ to also denote vectors in $\mathbb{R}^{TD}$, representing flattened versions of $\mathbf{s}_{1:T}$ and $\mathbf{r}_{1:T}$. We can therefore write the Jacobian of the residual for the whole sequence, $J(\mathbf{s})$, as a $TD \times TD$ matrix with block bidiagonal structure of the form

$$J(\mathbf{s}) := \frac{\partial \mathbf{r}}{\partial \mathbf{s}}(\mathbf{s}) = \begin{pmatrix} I_D & 0 & \dots & 0 & 0 \\ -\frac{\partial f}{\partial \mathbf{s}}(\mathbf{s}_1) & I_D & \dots & 0 & 0 \\ \vdots & \vdots & \ddots & \vdots & \vdots \\ 0 & 0 & \dots & I_D & 0 \\ 0 & 0 & \dots & -\frac{\partial f}{\partial \mathbf{s}}(\mathbf{s}_{T-1}) & I_D \end{pmatrix}. \tag{3}$$

## 3  DEER: Newton's Method for Parallel Evaluation of Sequential Models

Lim et al. [1] propose DEER, an algorithm using Newton's method for parallel evaluation of nonlinear sequential models, including both discrete-time nonlinear RNNs (GRUs, LSTMs, etc.) and neural ODEs [28, 29]. In this paper, we focus on the discrete-time setting, and address questions that arise from Lim et al. [1]: how to *scale* Newton's method, and how to make it *numerically stable*.

In this section we introduce DEER. We begin with a simplified derivation that emphasizes the link between Newton's method on vector spaces and parallelizable linear recurrences. We then present a new proof that DEER theoretically *always* converges globally. This proof also highlights why global convergence can be numerically unstable and/or slow in practice. We conclude by using these insights to discuss the weaknesses of DEER, and to motivate the methods we develop in Section 4.

### 3.1  Derivation of DEER from Newton's Method

The original derivation of DEER used Newton's method on Banach spaces and the Fréchet derivative for continuous-time systems to derive the update [1]. We specialize to the setting of discrete-time RNNs, and present a streamlined derivation that more directly connects the structure of the Jacobian in (3) to the linear recurrence relation in (6). This connection highlights why DEER incurs cubic work in $D$ and may encounter numerical instabilities. We will also use this form to prove DEER's global convergence in Section 3.2.

The $i^{\text{th}}$ Newton iterate for (2), starting at $\mathbf{s}^{(i)}$, is given by

$$\mathbf{s}^{(i+1)} \leftarrow \mathbf{s}^{(i)} - J(\mathbf{s}^{(i)})^{-1}\, \mathbf{r}(\mathbf{s}^{(\mathbf{i})}), \tag{4}$$

or equivalently,

$$\Delta \mathbf{s}^{(i+1)} := \mathbf{s}^{(i+1)} - \mathbf{s}^{(i)} = -J(\mathbf{s}^{(i)})^{-1}\, \mathbf{r}(\mathbf{s}^{(i)}). \tag{5}$$

Note this uses the root-finding view of Newton's method, see Appendix C.2.

The Jacobian defined in (3) is invertible and all of the eigenvalues are equal to one.[1] Storing and naively inverting the Jacobian is infeasible for large $D$ or $T$. However, since $J(\mathbf{s})$ is block bidiagonal, we can solve for $\Delta \mathbf{s}$ in (5) by forward substitution. This reduces to a simple recursion with the initial condition $\Delta \mathbf{s}_1^{(i+1)} = -\mathbf{r}_1(\mathbf{s}^{(i)})$, and for $t > 1$,

$$\Delta \mathbf{s}_t^{(i+1)} = \left[ \frac{\partial f}{\partial \mathbf{s}}(\mathbf{s}_{t-1}^{(i)}) \right] \Delta \mathbf{s}_{t-1}^{(i+1)} - \mathbf{r}_t(\mathbf{s}^{(i)}). \tag{6}$$

DEER uses the linearity of this recursion, solving it in parallel with a parallel associative scan [1, 5, 24]. Therefore, with $\mathcal{O}(T)$ processors, each Newton iteration can be performed in $\mathcal{O}(\log T)$ time.

We emphasize that the computation of the Newton step $\Delta \mathbf{s}$ in (5) is being parallelized. $J$ would, in general, be a $TD \times TD$ matrix that is prohibitive to store or invert. But by formulating this solve as an LDS in (6), we are able to parallelize the computation of $\Delta s$ (which consists of $T$ state updates, each of dimension $D$) over the sequence length. With sufficient processors, each update in (5) can be computed in $\mathcal{O}(\log T)$ time. Our implementation uses the parallel associative scan from JAX [30] (see Appendix B.6).

## 3.2 Global Convergence of DEER

We present a proof that DEER converges globally for discrete-time RNNs to the solution $\mathbf{s}_{1:T}^*$ of (2) in at most $T$ steps.

**Proposition 1.** *Undamped Newton's method will converge to the true solution, $\mathbf{s}_{1:T}^*$, of the fixed-point equation* (2) *in at most $T$ Newton iterations, for any initial $\mathbf{s}_{1:T}^{(0)}$.*

*Proof sketch.* For the full proof by induction, see Appendix A.1. The structure of $J(\mathbf{s})$ determines the recurrence in (6). The update applied at time $t$, $\Delta \mathbf{s}_t^{(i+1)}$, from (6) is the summation of a linearized $f$ applied to the update at time $t - 1$, and the residual one-step error at time $t$. Therefore, if the previous timestep is correct (i.e. $\Delta \mathbf{s}_{t-1}^{(i+1)} = \mathbf{0}$), then the update at time $t$ is just the one-step residual, which is defined exactly as the error. Therefore, if the previous value is correct, the updated current value will be correct. Given that $f$ and $\mathbf{s}_0$ are fixed and known, the result follows that all $T$ timesteps will have zero residual after $T$ iterations. □

It is not immediately obvious from (4) or past work Lim et al. [1] that DEER converges globally, but Proposition 1 shows that it does, at least theoretically. This result has two crucial corollaries. First, after $i$ Newton iterations, $\mathbf{s}_{1:T}^{(i)}$ will have zero error for all $t \leq i$. Therefore, if the iteration encounters numerical instabilities, as Newton is prone to, we can simply use a heuristic of resetting $s_t^{(i)}$ to a finite value for all $t > i$. This preserves the solution for time indices $t \leq i$ and allows the optimization to continue, but it is equivalent to running Newton's method from scratch on $\mathbf{s}_{i:T}$. This process is repeated until the entire trace has zero residual. A second corollary is that *any* set of finite matrices can replace $\{\partial f / \partial \mathbf{s}\}_{t=2}^T$ in (3) or (6), and the resulting quasi-Newton method will still converge globally in at most $T$ iterations. This preservation of global convergence provides further motivation for exploring quasi-Newton methods, as we discuss in the next section.

## 3.3 Weaknesses of DEER

Despite the theoretical convergence of DEER, its formulation as a linear recurrence relation in (6) highlights its limited scalability and stability. Scalability is limited because, in general, $\partial f / \partial \mathbf{s}$ is a dense $D \times D$ matrix. Therefore the parallel associative scan, which uses matrix-matrix multiplications, has $\mathcal{O}(TD^3)$ computational work and $\mathcal{O}(TD^2)$ memory complexity. Stability is limited because we often have no control over the eigenvalues of $\partial f / \partial \mathbf{s}$. If sufficiently many of these eigenvalues over the sequence length are larger in magnitude than one, then the linear recurrence relation will be numerically unstable. The heuristic approach of resetting unstable values is sufficient to ensure global convergence, but as we show in Section 6.3, it comes at the cost of runtime, as convergence is dramatically slower. These weaknesses motivate the development of two new techniques for parallel evaluation of RNNs: quasi-DEER and ELK, which we discuss in the next section.

---

[1]To see this fact, note that the characteristic polynomial of $J(\mathbf{s})$ in (3) is $(\lambda - 1)^{TD}$.

# 4 Scaling and Stabilizing Newton's Method for Parallel Evaluation

In Section 4.1 we introduce quasi-DEER, a quasi-Newton method that addresses the intractability of DEER for large state sizes. In Section 4.2 we introduce Evaluating Levenberg-Marquardt with Kalman (ELK), a damped Newton method for numerically stable, parallel evaluation of nonlinear RNNs. We also introduce quasi-ELK, which combines quasi-DEER and ELK to create a damped Newton's method for parallel sequential evaluation that is scalable and numerically stable.

## 4.1 Quasi-DEER: Scaling DEER with Diagonal Jacobian Approximations

As a consequence of our Proposition 1, replacing the Jacobians $\{\partial f/\partial s\}_{t=2}^{T}$ with an arbitrary matrix will still result in global convergence of the resulting DEER-like algorithm in at most $T$ iterations. A straightforward way to reduce the computational cost is to replace $\{\partial f/\partial s\}_{t=2}^{T}$ with $\{\text{diag}\left(\partial f/\partial s\right)\}_{t=2}^{T}$, i.e. take the diagonal entries of the Jacobians of the dynamics functions. The resulting linear recursion requires only $\mathcal{O}(TD)$ memory because it only needs to store diagonal matrices, and $\mathcal{O}(TD)$ work, because the parallelized associative scan only uses element-wise vector multiplies. Position-wise matrix-vector multiplies are still required to obtain the residuals, but this computation can be naively parallelized across the sequence.

Quasi-Newton methods approximate the Jacobian for computational reasons, so we refer to this algorithm as quasi-DEER. In Section 6, we show that quasi-DEER outperforms DEER on wall-clock time and memory usage on the tests from Lim et al. [1]. Quasi-DEER improves the scalability of DEER, but it does not address stability concerns. We propose a more stable solution below.

## 4.2 ELK: Stabilizing DEER with Trust Regions

Rather than treating RNN evaluation as a fixed-point finding problem, let us instead consider it as an optimization problem. First, we define the *merit function*

$$\mathcal{L}(\mathbf{s}) := \frac{1}{2} \left\| \mathbf{r}(\mathbf{s}) \right\|_2^2. \tag{7}$$

As in the fixed-point formulation, the unique minimizer of this objective is $\mathbf{s}^*$. In fact, the only local minimum of the merit function (7) is $\mathbf{s}^*$, as proved in Proposition 3 in Appendix A.2. One way of minimizing this *nonlinear* sum of squares objective is via the Gauss-Newton algorithm [23], which alternates between linearizing the terms in the merit function and solving the resulting *linear* sum-of-squares problem. The linearized objective at iteration $i$ is

$$\widetilde{\mathcal{L}}_{\mathbf{s}^{(i)}}(\Delta \mathbf{s}) = \frac{1}{2} \left\| \mathbf{r}(\mathbf{s}^{(i)}) + J(\mathbf{s}^{(i)})\Delta \mathbf{s} \right\|_2^2. \tag{8}$$

The solution is $\Delta \mathbf{s}^{(i+1)} = -J(\mathbf{s}^{(i)})^{-1}\, \mathbf{r}(\mathbf{s}^{(i)})$, which is exactly the DEER update from (5).

Formulating evaluation as nonlinear least squares also suggests more stable algorithms. The Levenberg-Marquardt algorithm [23] uses updates that solve a constrained optimization problem

$$\min_{\Delta \mathbf{s}} \widetilde{\mathcal{L}}_{\mathbf{s}^{(i)}}(\Delta \mathbf{s}) \quad \text{subject to } \left\| \Delta \mathbf{s} \right\|_2 \leq D_{i+1}, \tag{9}$$

where $D_{i+1}$ is an upper bound on the step size. We recognize this constraint as a trust region, which is often used in conjunction with Newton's method to improve numerical stability and convergence.

Finally, minimizing this constrained optimization is equivalent to minimizing the Lagrangian

$$\widehat{\mathcal{L}}(\Delta \mathbf{s}, \lambda_{i+1}) = \widetilde{\mathcal{L}}_{\mathbf{s}^{(i)}}(\Delta \mathbf{s}) + \frac{\lambda_{i+1}}{2} \left\| \Delta \mathbf{s} \right\|_2^2 \tag{10}$$

for some $\lambda_{i+1} \geq 0$. As noted by Särkkä and Svensson [25], the minimizer of this Lagrangian can be obtained by a Kalman smoother. We emphasize this connection in the following proposition.

**Proposition 2.** *Solving for the Levenberg-Marquardt update that minimizes* (10) *with fixed* $\lambda_{i+1}$ *is equivalent to finding the* maximum a posteriori *(MAP) estimate of* $\mathbf{s}_{1:T}$ *in a linear Gaussian state space model, which can be done in* $\mathcal{O}(\log T)$ *time on a sufficiently large parallel machine.*

*Proof.* Expanding the residual and Jacobian functions in (8), we see that up to an additive constant, the negative Lagrangian can be rewritten as,

$$- \widehat{\mathcal{L}}(\Delta \mathbf{s}, \lambda_{i+1}) \doteq \log \mathcal{N}\left(\mathbf{s}_1 \mid f(\mathbf{s}_0), I_D\right) + \sum_{t=1}^{T} \log \mathcal{N}\left(\mathbf{s}_t^{(i)} \mid \mathbf{s}_t, \frac{1}{\lambda_{i+1}} I_D\right)$$

$$+ \sum_{t=2}^{T} \log \mathcal{N}\left(\mathbf{s}_t \mid f(\mathbf{s}_{t-1}^{(i)}) + \left[\frac{\partial f}{\partial \mathbf{s}}(\mathbf{s}_{t-1}^{(i)})\right](\mathbf{s}_{t-1} - \mathbf{s}_{t-1}^{(i)}), I_D\right), \quad (11)$$

where $\mathcal{N}(\mathbf{x} \mid \boldsymbol{\mu}, \Sigma)$ denotes the probability density function of the multivariate normal distribution.

We recognize (11) as the log joint probability of a linear Gaussian state space model (LGSSM) [31] on $(\mathbf{s}_1, \dots, \mathbf{s}_T)$. The means of the dynamics distributions are given by the linearization of $f$, and the emissions are the previous iteration's states, $\mathbf{s}^{(i)}$. The parameter $\lambda_{i+1}$ sets the precision of the emissions, governing how far the posterior mode deviates from the previous states.

The minimizer of (10) is the posterior mode of the LGSSM (11), and can be obtained by Kalman smoothing [31]. As with the linear recursions in DEER, the Kalman smoother can be implemented as a parallel scan that scales as $\mathcal{O}(\log T)$ in time on a machine with $\mathcal{O}(T)$ processors [26, 27]. $\quad\square$

Therefore, we can evaluate an RNN by minimizing the merit function with the Levenberg-Marquardt algorithm. Since each step of the algorithm is performed by parallel Kalman smoothing, we call this approach *Evaluating Levenberg-Marquardt with Kalman* (ELK). Note that DEER is a special case of ELK, where $\lambda = 0$, which can be seen as minimizing the unpenalized linearized objective (8), or, alternatively, taking a Newton step with an infinitely large trust region. Moreover, under certain conditions, ELK also enjoys global convergence guarantees [23, Thms. 11.7, 11.8].

**Quasi-ELK: Scalability and Stability**    As with DEER, we can substitute an approximate Jacobian into the Lagrangian to obtain the *quasi-ELK* algorithm. Quasi-ELK enjoys the compute and memory scaling of quasi-DEER, as well as stability from the trust region damping from ELK. We show empirically in Section 6.3 that while quasi-ELK takes more iterates to converge than ELK, each quasi-ELK iterate is faster, giving overall runtime speedups.

**Implementation Details**    The convergence rate of (quasi-)ELK depends on the trust region radius $D_i$ (or alternatively $\lambda_i$). Although there exist methods to analytically set $\lambda_i$ [23, Algorithm 4.3], these approaches require factorizing $\partial\mathbf{r}/\partial\mathbf{s}$, which is intractable at scale. Therefore, we treat $\lambda$ as a hyperparameter set by a sweep over log-spaced values (cf. Appendix B.4).

We also use Kalman filtering instead of smoothing. We do so for two main reasons: filtering requires less work and memory; and we also found it to converge in fewer Newton iterations than smoothing. We hypothesize that this follows from Proposition 1, where the early part of the trace converges first. In Appendix A.3 we also discuss an alternative interpretation of ELK and the Kalman filter as defining a linear recurrence where the trust region attenuates the eigenvalues used in the parallel scan.

**Limitations**    The quasi-Newton methods lose the local quadratic convergence properties of Newton but remain globally convergent (cf. Proposition 1). Our implementation of quasi-DEER for training uses approximate gradients (cf. Section 6.2). The heuristic of resetting to the states to zero when they become unstable is also motivated by Proposition 1, but it slows convergence in (quasi-)DEER methods. As a result, we develop ELK to stabilize evaluation. Like DEER, ELK has cubic complexity in $D$, which we combat with quasi-ELK. However, quasi-ELK adds an additional hyperparameter. Note that all four parallelized methods discussed in this paper, as well as sequential evaluation of RNNs, have different regimes where they are fastest. For example, in our evaluation of autoregressive RNNs (Section 6.3), the ELK methods are faster than the DEER methods on wall-clock time, but they are slower than sequential. In our evaluation of the Lorenz96 system (Appendix B.5), ELK is more stable than DEER, but DEER is faster on wall-clock time. An area for future research is characterizing the properties of dynamical systems and hardware where each method is fastest. Finally, at the core of the implementation of the parallelized methods is the parallel associative scan (cf. Appendix B.6), which is currently available in JAX [30] but not in the standard PyTorch package.

# 5 Related Work

**RNNs and Parallelism**   Nonlinear RNNs are a natural choice for modeling sequential data because of their inductive biases and memory efficiency. However, most nonlinear RNNs are not parallelizable over the sequence length, and architectures that can exploit parallel computational hardware have been core to the success of deep learning. Therefore, a range of sequence architectures that inherently admit parallelism have been proposed, including transformers [3], deep linear RNNs [4–8, 32–35], and convolutions [36–39]. These methods obtain parallelism by developing new architectures, and do not consider parallelizing existing nonlinear architectures. DEER [1] is notable as it considers parallel evaluation and training of arbitrary nonlinear RNNs.

**Root Finding in Deep Learning**   Beyond DEER, there has been broad interest in using root finders/fixed-point iterations in deep learning and sequence modeling. Deep implicit layers [40–43] and neural ODEs [28, 44] replace conventional feed forward network layers [45] with an implicit layer whose output is the root of an equation. Moreover, Song et al. [46] parallelize the evaluation of feedforward nets using Jacobi and Gauss-Siedel iterations. In sequence modeling, parallel decoding methods [47–49] adapt ideas from Jacobi and Gauss-Siedel iterations to evaluate autoregressive sequence models in parallel. These approaches iteratively refine inputs by repeatedly feeding in previous outputs back into a parallelized sequence model. However, these methods presuppose the existence of a parallelized forward pass for the sequence model and do not leverage additional gradient information to obtain sublinear convergence.

**Parallelizing Dynamical Systems over Time**   Other work has investigated evaluating other nonlinear dynamical systems over time. ParaDIGMS [50] parallelizes sampling from diffusion models but uses Picard iterations instead of Newton's method, while Selvam et al. [51] use Parareal iterations. In the numerical ODE and PDE communities there has been great interest in Parallel in Time methods; see Gander [52], Ong and Schroder [53] for surveys. Vargas et al. [54] parallelized the evaluation of chaotic dynamical systems over time, but instead of casting Newton's method as a parallel scan, they resort to multi-grid methods to evaluate at different hierarchies. Moreover, these methods have not yet been applied to parallelizing RNNs.

**Scaling and Stabilizing Newton's Method**   Quasi-Newton methods are efficient algorithms that use an approximation of the Jacobian or Hessian in Newton's method, and include approaches like BFGS [55–58] and L-BFGS [59]. Other approaches use Newton's method to optimize deep nets [60]. However, these quasi-Newton algorithms do not admit efficient parallel scans. There are also conjugate gradients methods for exploiting structured Jacobians or Hessians [61], though they often do not attain the fast convergence rates of Newton or quasi-Newton methods [23]. Methods for stabilizing and ensuring Newton's method converges globally include regularization approaches [62, 63], backtracking line search [64], and trust regions [65]. All these stabilization methods have strengths and weaknesses, but as noted by Nocedal and Wright [23]: *"the trust-region Newton method has proved to be highly effective in practice,"* leading us to apply it to evaluating RNNs.

**Nonlinear Least Squares and Kalman Smoothing**   Bell and Cathey [66] and Bell [67] draw connections between the Gauss-Newton method and the iterated extended Kalman filter and smoother [68, 31]. Because Gauss-Newton is unstable, it is natural to use Levenberg-Marquardt [69, 70] to stabilize the filtering/smoothing problem [25, 71, 72]. These approaches start with a smoothing problem and stabilize it using approaches from nonlinear equations, whereas we start with a nonlinear equation to solve and make the connection with Kalman filtering to leverage parallelized algorithms [26]. We also discuss the practicalities of this connection for modern deep networks.

# 6 Experiments

We now experimentally examine the relative performance of these methods. Specifically, we evaluate: 1) whether quasi-DEER can provide memory savings over DEER and runtime savings over sequential evaluation, while retaining the accuracy of training and evaluation; and 2) whether ELK and quasi-ELK can be used to stabilize evaluation in regimes where DEER is unstable. In Sections 6.1 and 6.2 we use experimental designs from Lim et al. [1] and show that quasi-DEER retains

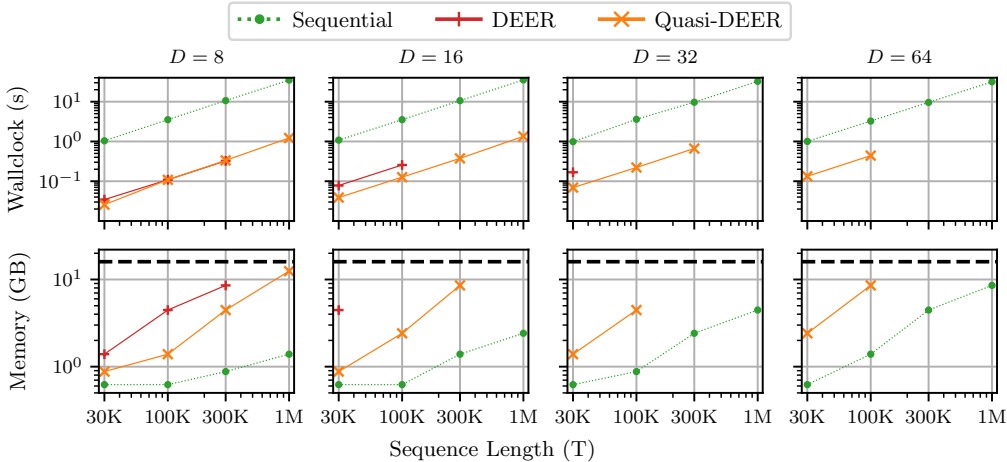

**Figure 2: Evaluating an untrained GRU.** Relative performance of sequential, DEER and quasi-DEER for evaluating a randomly initialized (and untrained) GRU on (**Top Row**) wall-clock time, averaged over 20 random seeds and (**Bottom Row**) memory, averaged over 3 random seeds. All experiments use a 16GB V100 SMX2 (memory capacity indicated by the black dashed line) and Newton methods were run to convergence. Missing points in each series indicate the GPU ran out of memory. Quasi-DEER has a runtime commensurate with DEER, but with lower memory consumption, allowing quasi-DEER to work at scales where DEER cannot. The accuracy of the final converged solution is similar for all methods (see Figure 5 in Appendix B.1).

the fast runtime and accuracy of DEER and can reduce memory consumption by up to an order of magnitude. In Section 6.3 we show that (quasi-)ELK remains stable when DEER becomes unstable, and that quasi-ELK is the fastest of all parallelized methods. We provide further details in Appendix B.

### 6.1 Quasi-DEER for Evaluation

We first use an experimental design from Lim et al. [1]. The task is to evaluate an untrained GRU across a range of hidden state sizes ($D$) and sequence lengths ($T$) on a 16GB V100 GPU; the inputs to the RNN also have dimension $D$. We compare the wall-clock time and memory usage of three methods: sequential evaluation, DEER, and quasi-DEER. Results are shown Figure 2.

Both DEER and quasi-DEER are up to twenty times faster than sequential evaluation. The runtimes are similar between DEER and quasi-DEER for small networks, because although quasi-DEER steps are faster, quasi-DEER takes more iterations to converge. For larger networks, the difference in runtime is more pronounced. We also see that quasi-DEER requires as much as an order of magnitude less memory than DEER, thus allowing the application to architectural regimes previously infeasible with DEER. In Figure 6 of Appendix B.1.1 we show that in smaller $T$ and $D$ regimes we observe the expected sublinear time scaling with sequence length. This experiment confirms that quasi-DEER can replicate the performance of DEER, but with a smaller memory footprint.

### 6.2 Quasi-DEER for Training

We verify that quasi-DEER expedites training nonlinear RNN models. We replicate the third experiment from Lim et al. [1], where a GRU is trained to classify *C. elegans* phenotypes from the time series of principal components of the worms' body posture [73].

We show results in Figure 3. We see that the training dynamics under quasi-DEER leads to the similar validation accuracy trajectories. However, every quasi-DEER training step is faster by a factor of 2.5, despite performing around 2 times more Newton iterations per training step. This finding highlights how quasi-DEER can replace DEER when training nonlinear RNNs, yielding both time and memory savings. In our experiment, we use the quasi-DEER approximation for the backward pass as well, leading to gradients that are different from DEER in this setting. However, we find that there is negligible degradation in performance (Figure 3, left).

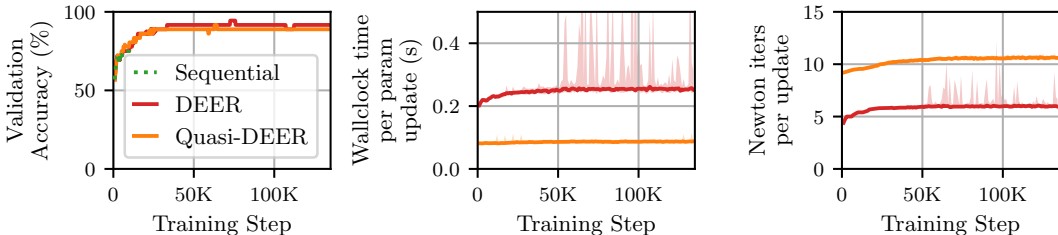

**Figure 3: Training a GRU with DEER.** Comparison of DEER and quasi-DEER during GRU training for the *C. elegans* time-series classification task (Section 6.2). Each time series has length $T = 17,984$. We show the median, and 5-95% interval across a rolling window of 20 training steps. **(Left)** DEER and quasi-DEER have the similar validation accuracy trajectories, indicating similar training dynamics. The sequential trace shown is for 24 hours of training (compared to 11 and 4 hours for the whole DEER and quasi-DEER traces). **(Center)** Each quasi training iteration is 2.5 times faster than each DEER training iteration. Sequential training steps took more than 6 seconds each (not pictured). **(Right)** Each quasi training iteration requires (approximately) 2 times more Newton iterations to converge, indicating that each quasi Newton step is approximately 5 times faster than the corresponding DEER Newton step.

DEER is prone to "spikes", where orders of magnitude more steps are required for convergence (Figure 3, right). While quasi-DEER is not as susceptible to these spikes (never more than half an order of magnitude), these instabilities motivated the study of stabilizing methods.

### 6.3 ELK and Quasi-ELK for Evaluating Autoregressive RNNs

We conclude by studying an application where these numerical instabilities in DEER are critical. We use a small autoregressive GRU (hidden dimension $N_h = 3$), where the previous sampled value is input into the GRU at the next step. Such autoregressive architectures were not examined by Lim et al. [1], but are an important class of models. We describe the precise details of the AR GRU we use in Appendix B.3. Crucially, the Markovian state $\mathbf{s}_t$ used by all four parallelized methods must be expanded to include the current sampled output value, as well as the current GRU state.

**Initialized AR GRU** We first repeat the analysis in Section 6.1 (and similar to the evaluation in Lim et al. [1]) for evaluating a randomly initialized autoregressive GRU. We see in the top left panel of Figure 4 that all four parallelized methods converge rapidly and stably to the correct trace, indicated by a low mean absolute discrepancy (MAD) between the true trace and the generated trace.

**Trained AR GRU** We then study a pre-trained GRU that generates a noisy sine wave (see Figure 4, bottom). The linear recurrence relation (6) was numerically unstable in DEER and quasi-DEER. To remedy these instabilities, we take the approach described earlier of setting the unstable parts of the trace to a fixed value (here zero). Doing so ensures convergence, but at the cost of "resetting" the optimization for large swathes of the trace (Figure 4, bottom) and slowing convergence (see Figure 4, top right). This finding highlights how the instabilities of DEER — which are inherited from both pathologies of Newton's method and the parallel recurrence — can be crippling in even very simple scenarios. While resetting allows for convergence, the resulting convergence is very slow.

We then apply ELK and quasi-ELK. We show the results in the top right and bottom panels of Figure 4. We select the trust region size with a one-dimensional search over log-spaced values between $10^0$ and $10^7$. We see ELK has stabilized convergence, with the evaluation never incurring numerical instabilities or requiring heuristics. Crucially, by taking more stable steps (and not needing stabilizing heuristics) ELK and quasi-ELK converge faster than DEER and quasi-DEER. ELK can stabilize and expedite the convergence of DEER, with quasi-ELK faster still (by wall-clock time).

However, on this task, all parallelized methods (including DEER) are slower than sequential generation. Quasi-ELK is the fastest parallel method, taking 221 milliseconds, compared to sequential evaluation, taking 96 milliseconds. For comparison, DEER took 1,225 milliseconds. Quasi-ELK therefore still represents a large improvement in runtime over previous parallel methods. We provide timing details and further discussion in Appendix B.3.2.

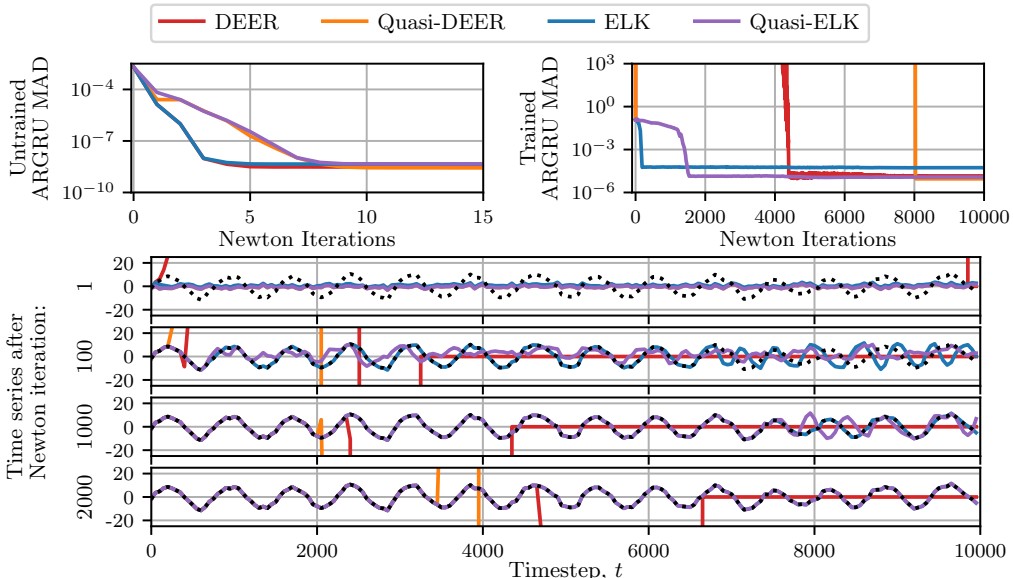

**Figure 4:** ELK stabilizes parallel evaluation of an AR GRU. **(Top Left)** The mean absolute difference (MAD) evaluated on the outputs converges rapidly for all four methods on a sequence generated by an *untrained* AR GRU. **(Top Right)** The MAD for evaluating a trained AR GRU. Undamped DEER variants are unstable and converge slowly (using the reset heuristic). ELK stabilizes and accelerates convergence. **(Bottom)** The output after 1, 100, 1000, and 2000 Newton iterations. The black dotted line is the true trace. ELK and quasi-ELK converge rapidly, but DEER and quasi-DEER are unstable. The lines where DEER and quasi-DEER are zero depict the zeroing heuristic.

## 7 Conclusion

In this paper we proposed methods for scalable and stable parallel evaluation of nonlinear RNNs. DEER [1] achieved speedups over sequential evaluation, but incurred quadratic memory, cubic work, and numerical instabilities. We therefore extended DEER to use quasi-Newton approximations that reduced computational complexity, and we provided a novel proof that both DEER and quasi-DEER converge globally. To stabilize DEER, we leveraged a connection between the Levenberg-Marquardt method and Kalman smoothing to enable parallel evaluation of RNNs, allowing us to stabilize DEER while still leveraging fast parallel filtering. We empirically verified that quasi-DEER, ELK, and quasi-ELK improve convergence across a range of metrics and examples. This result allows parallel evaluation of nonlinear RNNs to be scaled beyond what is possible with DEER.

When selecting an approach, we offer the following advice: If rapid convergence is reliably observed, our experiments show that quasi-DEER provides the fastest convergence in terms of wall-clock time. However, if the dynamics are on the edge of stability, then ELK offers the most stable performance of these parallelized methods, but quasi-ELK could be faster in wall-clock time and just as stable. In such settings, it is worth sweeping the hyperparameter to choose the best version of ELK (note that for $\lambda = 0$, ELK specializes to DEER). However, in the setting of chaotic dynamics, standard sequential evaluation may ultimately be faster.

Our experiments and these observations also highlight avenues for future research. While we found success with a diagonal approximation, structured approximations of the Jacobian that still admit fast parallelism but are more faithful approximations may allow for more accurate quasi-Newton steps to be taken. Secondly, quantifying the convergence rates of quasi-ELK would allow us to provide tighter bounds than those derived in Proposition 1. Finally, theoretically investigating whether further improvements to parallelized methods can prove faster than sequential evaluation for dynamical systems on the edge of stability, or whether there are fundamental limitations to the computational benefit of parallelization, are interesting questions for future work.

## Acknowledgments

We thank John Duchi, David Zoltowski, and the members of the Linderman Lab for their helpful feedback. We thank Amber Hu for her work on preliminary versions of this project, and Leo Kozachkov for pointing out that the conditions we established for the merit function are equivalent to invexity. We thank the anonymous NeurIPS reviewers whose feedback improved this paper.

This work was supported by grants from the NIH BRAIN Initiative (U19NS113201, R01NS131987, & RF1MH133778), the NSF/NIH CRCNS Program (R01NS130789). X.G. would also like to acknowledge support from the Walter Byers Graduate Scholarship from the NCAA. S.W.L. is supported by fellowships from the Simons Collaboration on the Global Brain, the Alfred P. Sloan Foundation, and the McKnight Foundation. The authors have no competing interests to declare.

Some of the experiments were performed on the Sherlock cluster. We thank Stanford University and the Stanford Research Computing Center for providing computational resources and support that contributed to these research results.

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

# A   Theoretical Results

## A.1   Proof of Proposition 1

**Proposition 1** *Undamped Newton's method will converge to the true solution, $\mathbf{s}^*$, of the fixed-point equation*(2) *in at most $T$ Newton iterations, for any initial $\mathbf{s}^{(0)}$.*

*Proof.*   We prove this result by induction on the sequence length.

In general, the guess $\mathbf{s}^{(0)}$ need not equal the solution $\mathbf{s}^*$ anywhere. However, the initial state $\mathbf{s}_0$ and the dynamics functions $f$ are fixed. Therefore, $\mathbf{s}_1^* = f(\mathbf{s}_0)$ and in general $\mathbf{s}_t^* = f(\mathbf{s}_{t-1}^*)$. Thus, it follows from the initial condition of the DEER recurrence relation that $\mathbf{s}_1^{(i)} = \mathbf{s}_1^*$ for all $i \geq 1$.

Furthermore, we observe that if $\mathbf{s}_t^{(i)} = \mathbf{s}_t^*$ for all $t$ less than some $t^{(i)}$, then $\mathbf{r}_t(\mathbf{s}^{(i)}) = \mathbf{0}$ for all $t < t^{(i)}$ by the definition of the residual in (1). Therefore, the DEER linear recurrence relation necessarily gives $\Delta\mathbf{s}_t^{(i+1)} = \mathbf{0}$ for all $t < t^{(i)}$. Furthermore, because $\mathbf{s}_{t^{(i)}}^* = f(\mathbf{s}_t^{(i)})$, it follows that $\Delta\mathbf{s}_{t^{(i)}}^{(i+1)} = -\mathbf{r}_{t^{(i)}}(\mathbf{s}^{(i)}) = \mathbf{s}_{t^{(i)}}^* - \mathbf{s}_{t^{(i)}}^{(i)}$. Thus, it follows that after applying another Newton iteration that $\mathbf{s}_t^{(i+1)} = \mathbf{s}_t^*$ for all $t < t^{(i)} + 1$. The global convergence result and bound on Newton iterates follows by induction. □

We note that this proof technique (induction) is very similar to that used to prove Proposition 1 of Shih et al. [50]. However, Shih et al. [50] proves a result about Picard iterations (zeroth order method). Our proof about the global convergence of Newton iterations contains the additional complication of dealing with a LDS (as a consequence of using a first order method). Note, that the global convergence comes from the zeroth order update; however, the first order gradient information can accelerate convergence. The fact that both Picard and Newton iterations have very similar proofs of global convergence suggests that they are closely related, and we will expand on this connection in future work.

**Discussion of corollaries of Proposition 1**   Corollaries of Proposition 1 include that the fixed-point iterations will converge (in at most $T$ iterations) to $\mathbf{s}^*$ even if the Jacobians $\partial f/\partial\mathbf{s}$ are replaced by *arbitrary* matrices, and that we can reset the values of $\mathbf{s}_t^{(i)}$ for $t > i$ arbitrarily and still enjoy convergence.

In more detail, the elements in the sub-block-diagonal of $J := \partial\mathbf{r}/\partial\mathbf{s}$ can be replaced with arbitrary values – but the main block diagonal must remain as the identity and all other entries must be zero. Retaining convergence under modifications to the sub-block-diagonal portion is a corollary of Proposition 1, and can be seen from (6): If all the states up to and including position $t - 1$ at the $(i)$th Newton iteration are correct, then the update in (6) at Newton iteration $(i + 1)$ for position $t$ will use $\Delta\mathbf{s}_{t-1}^{(i+1)} = \mathbf{0}$ (no update is required at position $t - 1$), and so the update to $\mathbf{s}_t^{(i+1)}$ no longer depends on the Jacobian.

We exploit this to develop quasi-DEER, retaining only the diagonal of the Jacobians. Doing so reduces the parallel scan from $\mathcal{O}(D^3)$ work to $\mathcal{O}(D)$ work, making each iteration faster (while still admitting global convergence as above), but needs more Newton iterations to converge due to approximate updates. We find that this trade-off often yields a faster wall-clock time (cf. Figure 8).

Explicitly, the global convergence of quasi-DEER is a theoretical result (a corollary of Proposition 1), but the fast runtime of quasi-DEER in practice is an empirical result (cf. Figure 3).

## A.2   The Merit Function Has No Local Minima or Saddle Points

**Proposition 3.** *The merit function $\mathcal{L}(\mathbf{s})$ defined in (7) has a global minimum at the true trace $\mathbf{s}^*$, satisfying $\mathcal{L}(\mathbf{s}^*) = 0$. It has no other critical points, i.e. no $\mathbf{s}$ such that $\nabla\mathcal{L}(\mathbf{s}) = \mathbf{0}$ other than at the unique $\mathbf{s}^*$ for which $\mathbf{r}(\mathbf{s}^*) = \mathbf{0}$.*

*Proof.*   First, we observe that $\nabla\mathcal{L}(\mathbf{s}) = J(\mathbf{s})^T\mathbf{r}(\mathbf{s})$, where $J(\mathbf{s})$ is defined as in (3). Because $J(\mathbf{s})$ is a lower triangular matrix with all entries on its diagonal equal to 1, it follows that all of its eigenvalues

are equal to 1. Therefore, $J(\mathbf{s})$ is nonsingular for all $\mathbf{s}$. Thus, $J(\mathbf{s})$ has trivial nullspace for all $\mathbf{s}$, i.e. $J(\mathbf{s})^T \mathbf{r}(\mathbf{s}) = \mathbf{0} \iff \mathbf{r}(\mathbf{s}) = \mathbf{0}$. But only $\mathbf{s}^*$ satisfies $\mathbf{r}(\mathbf{s}^*) = \mathbf{0}$. $\qquad\qquad\qquad\square$

Since there are no critical points other than $\mathbf{s}^*$, the merit function cannot have local minima or saddle points. Since we have shown that every stationary point is a global minimum, it follows that the merit function $\mathcal{L}(\mathbf{s})$ is *invex* (cf. [74, 75]).

We also discuss further the uniqueness of the global minimizer $\mathbf{s}^*$ of the merit function $\mathcal{L}$.

For a deterministic forward function $f$ and fixed inputs there is a fixed sequence of states and outputs (note that any stochastic dynamics function can be reparameterized as deterministic by conditioning on the random inputs). Thus, $\mathbf{s}^*$ is the only sequence with zero residual (i.e. there is a unique sequence generated by the deterministic dynamics).

Furthermore, DEER cannot get stuck at any point that is not this sequence. We prove this in Proposition 1. Another way to see this however is that each update step (4) is equal to $\mathbf{J}^{-1}\mathbf{r}$. But, $\mathbf{J}$ is always invertible and so has trivial nullspace. Furthermore, the residual $\mathbf{r}$ can only be zero at the unique solution $\mathbf{s}^*$. Thus $\mathbf{J}^{-1}\mathbf{r}$ is nonzero everywhere except at the true solution, where it is zero. Thus, DEER cannot get stuck en route to finding the true and unique solution.

### A.3 Kalman Filtering Damps the Eigenvalues of the Dynamics Matrices

A complementary perspective on how ELK results in more stable evaluation of nonlinear RNNs is to see how the Kalman filter damps the eigenvalues of the Jacobian matrices of the transition dynamics. We first provide a high-level overview, and then provide a more detailed derivation.

**Overview**  Let $\mathbf{A}_t$ be the Jacobians $\partial f / \partial \mathbf{s}$ used in the linear recurrence relations and $\mathbf{b}_t$ be the offsets. Then the prediction step of the Kalman filter (ELK) is the same as DEER. However, after applying the update step in ELK (which imposes the trust region), we obtain a second linear recurrence relation where the linear operator is given by $\mathbf{\Gamma}_t \mathbf{A}_T$. Note that $\mathbf{\Gamma}_t$ is a symmetric positive definite matrix with eigenvalues bounded above by $1/1+\lambda$. Thus, by the Spectral Theorem, it follows that the norms of the eigenvalues of $\mathbf{\Gamma}_t \mathbf{A}_t$ are bounded above by the max of the norms of the eigenvalues of $\mathbf{A}_t$, scaled by $1/1+\lambda$. Note that larger $\lambda$ corresponds to more regularization/smaller trust region; and therefore correspondingly results in smaller effective eigenvalues in the scan. We recover DEER exactly if $\lambda = 0$. Thus, while large eigenvalues in $A_t$ are the cause of the instability of DEER when evaluating unstable dynamical systems, ELK directly attenuates these large eigenvalues, explaining why the intermediate iterations using ELK remain stable.

**Derivation**  We define our dynamics used in Newton iteration $(i + 1)$ as

$$\mathbf{A}_t = \frac{\partial f_{t+1}}{\partial \mathbf{s}}(\mathbf{s}_t^{(i)})$$

$$\mathbf{b}_t = f_{t+1}(\mathbf{s}_t^{(i)}) - \frac{\partial f_{t+1}}{\partial \mathbf{s}}(\mathbf{s}_t^{(i)})\mathbf{s}_t^{(i)}.$$

Now $\mathbf{A}_t \in \mathbb{R}^{D \times D}$ and $\mathbf{b}_t \in \mathbb{R}^D$.

In line with considering the system as the LDS in (11), we set the process noise to be $\mathbf{I}_D$, and with the emissions governed by

$$\mathbf{s}_t^{(i+1)} \sim \mathcal{N}(\mathbf{s}_t^{(i)}, \sigma^2 \mathbf{I}_D),$$

where $\sigma^2$ controls the size of our trust region (note that in the notation of developed in Section 4.2 we have $\lambda = 1/\sigma^2$).

In the notation of Murphy [76], we see that the **predict step** is

$$\boldsymbol{\mu}_{(t+1)|t} = \mathbf{J}_t \boldsymbol{\mu}_{t|t} + \mathbf{b}_t$$

$$\mathbf{\Sigma}_{(t+1)|t} = \mathbf{A}_t \mathbf{\Sigma}_{t|t} \mathbf{A}_t^T + \mathbf{I}_D.$$

Meanwhile, the **update step** is

$$\boldsymbol{\mu}_{(t+1)|(t+1)} = \boldsymbol{\mu}_{(t+1)|t} + \mathbf{\Sigma}_{(t+1)|t}(\mathbf{\Sigma}_{(t+1)|t} + \sigma^2 \mathbf{I}_D)^{-1}(\mathbf{y}_{t+1} - \boldsymbol{\mu}_{(t+1)|t})$$

$$\mathbf{\Sigma}_{(t+1)|(t+1)} = \mathbf{\Sigma}_{(t+1)|t} - \mathbf{\Sigma}_{(t+1)|t}(\mathbf{\Sigma}_{(t+1|t)} + \sigma^2 \mathbf{I}_D)\mathbf{\Sigma}_{(t+1|t)}^T.$$

To unpack this further, we first define the *attenuation matrix*

$$\boldsymbol{\Gamma}_t = \sigma^2 \Big( \mathbf{A}_t \boldsymbol{\Sigma}_{t|t} \mathbf{A}_t^T + (\sigma^2 + 1)\mathbf{I}_D \Big)^{-1}.$$

Because $\boldsymbol{\Sigma}_{t|t}$ is a covariance matrix, it is also symmetric positive definite, and so $\mathbf{A}_t \boldsymbol{\Sigma}_{t|t} \mathbf{A}_t^T$ is symmetric positive definite, and so all of its eigenvalues are greater than zero. Therefore, all the eigenvalues of $\mathbf{A}_t \boldsymbol{\Sigma}_{t|t} \mathbf{A}_t^T + (\sigma^2 + 1)\mathbf{I}_D$ are greater than $\sigma^2 + 1$.

We note that $\boldsymbol{\Gamma}_t$ is also symmetric and positive definite. Thus, by the Spectral Theorem, all eigenvalues of $\boldsymbol{\Gamma}_t$ are positive. By the above argument, the eigenvalues of $\boldsymbol{\Gamma}_t$ are all less than $\frac{\sigma^2}{1+\sigma^2} < 1$.

Thus, we observe that the resulting filtering is given by the recurrence relation

$$\boldsymbol{\mu}_{(t+1)|(t+1)} = \overbrace{\boldsymbol{\Gamma}_t \mathbf{A}_t \boldsymbol{\mu}_{t|t}}^{\text{linear}} + \overbrace{\boldsymbol{\Gamma}_t \mathbf{b}_t + (\mathbf{A}_t \boldsymbol{\Sigma}_{t|t} \mathbf{A}_t^T + \mathbf{I}_D)\Big( \mathbf{A}_t \boldsymbol{\Sigma}_{t|t} \mathbf{A}_t^T + (\sigma^2 + 1)\mathbf{I}_D \Big)^{-1} \mathbf{y}_{t+1}}^{\text{offset}}$$

$$\boldsymbol{\Sigma}_{(t+1)|(t+1)} = \boldsymbol{\Gamma}_t (\mathbf{A}_t \boldsymbol{\Sigma}_{t|t} \mathbf{A}_t^T + \mathbf{I}_D).$$

Given $\left\{ \boldsymbol{\Sigma}_{t|t} \right\}_{t=0}^{T-1}$, we see that the filtered means (the updates for ELK) come from a linear recurrence relation with linear term $\boldsymbol{\Gamma}_t \mathbf{A}_t$.

We therefore compare the eigenvalues of $\boldsymbol{\Gamma}_t \mathbf{A}_t$ to eigenvalues of $\mathbf{A}_t$. Because $\boldsymbol{\Gamma}_t$ is symmetric positive definite, by the Spectral Theorem we can write $\boldsymbol{\Gamma}_t = \mathbf{Q} \boldsymbol{\Lambda}_t \mathbf{Q}^T$, where $\mathbf{Q}$ is an orthogonal (and therefore unitary) matrix, and $\boldsymbol{\Lambda}_t$ is a diagonal matrix where every entry is in $(0, 1)$ (the entries of $\boldsymbol{\Lambda}_t$ are the eigenvalues of $\boldsymbol{\Gamma}_t$, which are greater than $0$ by the Spectral Theorem and less than $\frac{\sigma^2}{1+\sigma^2} < 1$ by the argument above).

Now, let's consider any arbitrary unit vector $\mathbf{v} \in \mathbb{C}^D$, and let $\boldsymbol{\Lambda}_t^{\max}$ denote the maximum of the norms of all eigenvalues of $\mathbf{A}_t$. Then $\|\mathbf{A}_t \mathbf{v}\|_2 \leq \boldsymbol{\Lambda}_t^{\max}$ by the definition of $\boldsymbol{\Lambda}_t^{\max}$. However, we want to know $\|\mathbf{Q}\boldsymbol{\Lambda}_t\mathbf{Q}^T\mathbf{A}_t\mathbf{v}\|_2$ for any arbitrary unit vector $\mathbf{v} \in \mathbb{R}^D$. However, we know that the action of a unitary matrix cannot change the 2-norm of a vector, so $\|\mathbf{Q}\boldsymbol{\Lambda}_t\mathbf{Q}^T\mathbf{A}_t\mathbf{v}\|_2 = \|\boldsymbol{\Lambda}_t\mathbf{Q}^T\mathbf{A}_t\mathbf{v}\|_2$. Moreover, multiplying a vector by a diagonal matrix cannot increase the 2-norm of a vector by more than the absolute value of the diagonal matrix, which in the case of $\boldsymbol{\Lambda}_t$ is bounded above by $\sigma^2/\sigma^2+1$. Thus, $\|\mathbf{Q}\boldsymbol{\Lambda}_t\mathbf{Q}^T\mathbf{A}_t\mathbf{v}\|_2 \leq \frac{\sigma^2}{\sigma^2+1}\|\mathbf{A}_t\mathbf{v}\|$, or

$$\|\mathbf{Q}\boldsymbol{\Lambda}_t\mathbf{Q}^T\mathbf{A}_t\mathbf{v}\|_2 \leq \frac{\sigma^2}{1+\sigma^2}\boldsymbol{\Lambda}_t^{\max}$$

for any unit vector $\mathbf{v} \in \mathbb{C}^D$. This highlights that we can interpret reducing $\sigma^2$ (reducing the size of the trust region and increasing stabilization) as directly attenuating the eigenvalues in the linear recurrence, helping to combat eigenvalues with large magnitude.

### A.4   Scale-ELK

Motivated by our derivation in Appendix A.3, which shows that ELK reduces the magnitude of the eigenvalues of the Jacobian matrices in the transition dynamics, we recommend a more lightweight version of ELK which we call *scale-ELK*.

Scale-ELK uses a hyperparameter $k \in [0, 1]$ (as opposed to $\lambda \in [0, \infty)$ used by ELK). Scale-ELK uses a linear dynamical system just like DEER, with the dynamics defined as

$$\mathbf{A}_t = (1 - k)\frac{\partial f_{t+1}}{\partial \mathbf{s}}(\mathbf{s}_t^{(i)})$$

$$\mathbf{b}_t = f_{t+1}(\mathbf{s}_t^{(i)}) - (1 - k)\frac{\partial f_{t+1}}{\partial \mathbf{s}}(\mathbf{s}_t^{(i)})\mathbf{s}_t^{(i)}.$$

Thus, setting $k = 0$ recovers DEER, while setting $k = 1$ recovers a (computationally expensive form of) sequential evaluation. Ideally, $k$ is chosen to keep the magnitude of the eigenvalues of $\{A_t\}_{t=1}^T$ below 1. By Proposition 1, scale-ELK also enjoys global convergence.

Scale-ELK enjoys two primary benefits over ELK. First, an evaluation of scale-ELK uses fewer FLOPs than ELK, as scale-ELK is just parallelizing an LDS with ELK uses a parallelized Kalman filter. Second, the Kalman filter involves inverses which run the risk of introducing numerical instability, while scale-ELK avoids these complications.

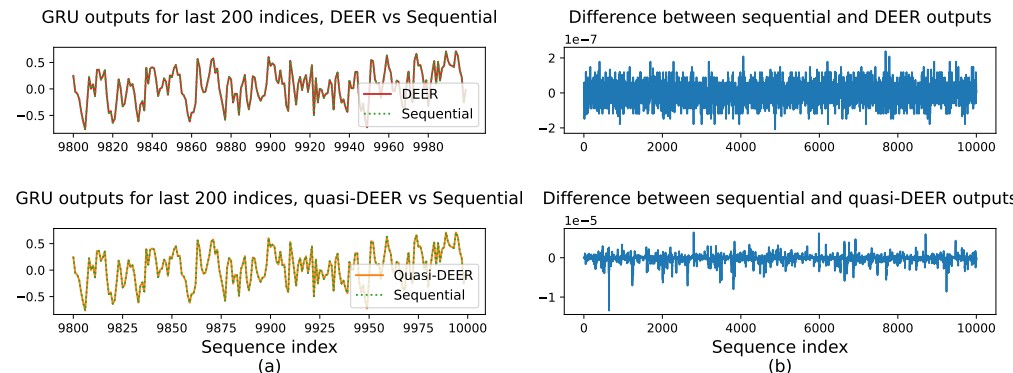

**Figure 5:** The accuracy of evaluating with parallelized methods (DEER and quasi-DEER) as opposed to sequential evaluation. The parallelized methods converge to the correct trace within numerical precision. The hidden state size is $D = 4$ and the sequence length is $T = 10,000$.

## B    Experimental Details

### B.1    Quasi-DEER for Evaluation

In this section we elaborate on our Experiment 1, discussed in Section 6.1. We closely follow the experimental design in Section 4.1 of Lim et al. [1], including 5 warm-up steps for all timing experiments and a batch size of 16. However, instead of running 5 seeds for 20 repetitions each, we run 20 seeds for 5 repetitions each, to get more coverage over different evaluation (as the timing for each evaluation are observed to be low variance). We also include memory profiling experiments not present in Lim et al. [1]. For these experiments we use 3 random seeds and only one repetition, because the memory usage is extremely stable in between runs. We discus the memory profiling experiments in more detail in Section B.1.3.

For runs with the same specifications (sequence length $T$, hidden state size $D$, and algorithm), we observe that sometimes runs with memory profiling ran out of memory whereas runs with timing profiling did not run out of memory. A difference between our time profiling runs and memory profiling runs was how we handled preallocation of memory. For time profiling, we allowed JAX to preallocate memory because that is how JAX usually runs, and so gives a better indication of wall-clock time in practice. For memory profiling, we did not allow JAX to preallocate memory so we could get a more fine-grained measure of memory usage.

However, JAX provides this following discussion of memory preallocation (see `https://jax.readthedocs.io/en/latest/gpu_memory_allocation.html#`):

> However, [not preallocating memory] is more prone to GPU memory fragmentation, meaning a JAX program that uses most of the available GPU memory may OOM with preallocation disabled.

Because the specifications where the time profile runs stay within memory but the memory runs run out of memory are likely very close to the 16GB threshold, our hypothesis is that this phenomenon is a manifestation of this documented memory fragmentation.

### B.1.1    Numerical Precision of DEER and Quasi-DEER

In Figure 5 we qualitatively show that for the same example used in Figure 3 of Lim et al. [1] that quasi-DEER converges within numerical precision to the correct trace in the untrained GRU benchmarking task discussed in Section 6.1. Similar results for DEER can be found in Section 4.1 of Lim et al. [1].

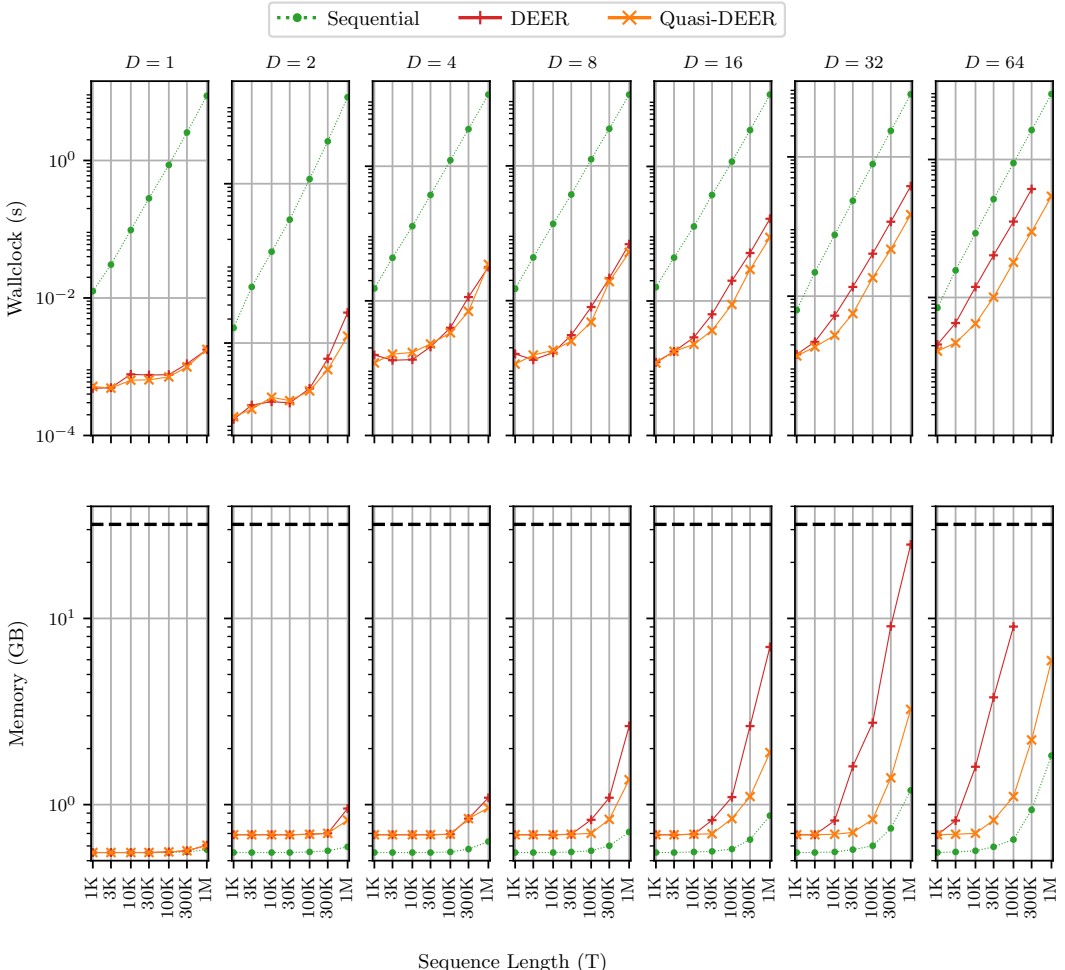

**Figure 6: Evaluating an untrained GRU.** Sublinear and linear timing regimes for parallelized algorithms. The above experiments were run on a 32 GB V100 with a batch size of 1. As in Figure 2, we use 20 seeds for timing, 3 seeds for memory, and the dashed black line indicates the memory capacity of the GPU (32 GB). We observe that in smaller regimes in $D$ and $T$ that the wall-clock time shows sublinear scaling indicative of the use of parallel algorithms. However, when the GPU becomes saturated, the benefits of parallelization are reduced and we begin to see linear scaling in wall-clock time with $T$.

### B.1.2 Different Scaling Regimes Depending on GPU Saturation

In Figure 6, we run the timing benchmarks of Section 6.1 on a wider range of sequence lengths $T$ and hidden state sizes $D$, on a larger GPU (a V100 with 32 GB) and with a smaller batch size of 1. In doing so, we highlight that the parallel nature of DEER and quasi-DEER, as their wall-clock time scales sublinearly in the sequence length $T$ in smaller $(D, T)$ regimes. However, we note that in the larger regimes considered in our main text and in Lim et al. [1], we often observe linear scaling in the sequence length $T$ for the wall-clock time of DEER and quasi-DEER, even though these algorithms are still faster than sequential evaluation. Figure 6 shows good evidence that these parallel algorithms are suffering from saturation of the GPU, and would benefit from even more optimized parallel hardware

The parallel scan, given sufficiently many processors, scales as $O(\log T)$. As we show in Figure 6, we see this speedup at low model sizes and sequence lengths. Once the processors are saturated, we see a linear increase in the runtime (since the amount of work done is linear), but it is making much more effective use of the GPU, resulting in a constant factor speedup over sequential application at larger model sizes/sequence lengths.

### B.1.3 Memory Profiling Details

As we discussed in Section 4.1, quasi-DEER is $\mathcal{O}(TD)$ in memory while DEER is $\mathcal{O}(TD^2)$ in memory because DEER uses dense Jacobians $\partial f/\partial s$ while quasi-DEER uses a diagonal approximation, $\mathrm{diag}(\partial f/\partial s)$. However, to implement quasi-DEER with automatic differentiation, the most standard approach would be to compute the dense Jacobian, and then to take the diagonal; however, such an approach would still be $\mathcal{O}(TD^2)$ in memory required. There are two implementation workarounds. One is to loop over computing partial derivatives, effectively trading time for memory. The second is simply derive the diagonal entries of the Jacobian for the architecture of interest. For the purpose of showcasing the $\mathcal{O}(TD)$ memory usage of quasi-DEER in Section 6.1, we take this second approach, deriving the diagonal entries of the Jacobian of the GRU nonlinear dynamics and implementing them in JAX. However, for our other experiments, where memory capacity is not a problem, we simply use the less memory efficient version of quasi-DEER.

We also see linear memory scaling in evaluating the RNN sequentially. This behavior occurs because we JIT compile a `lax.scan` in JAX, and we track the maximum memory used on the GPU at any point in the computation. Because the inputs and the hidden states of the RNN scales are both of length $T$, the memory usage of $\mathcal{O}(T)$. While there may be more memory efficient ways to sequentially evaluate an RNN, we keep the same benchmarking structure as Lim et al. [1] for to make comparison easier.

## B.2 Quasi-DEER for Training

Here we discuss the experimental details for Experiment 2 in Section 6.2. We follow the same experimental setup as in Section 4.3 and Appendix B.3 of Lim et al. [1]. As an aside, we note that the choice of hardware can impact behavior of the algorithms dramatically. For replicability, we run on the same hardware as Lim et al. [1], using a 16GB V100 SXM2. However, we note that if we try to run these same experiments on A100, DEER struggles to converge numerically, although quasi-DEER shows no such difficulty. If we run on a CPU, both DEER and quasi-DEER converge numerically. On balance, on the eigenworms time series classification task, both DEER and quasi-DEER are numerically stable for the most part; the numerical instabilities we have observed for DEER on an A100 are likely specific to some particular detail of JAX/hardware interaction.

In our implementation of quasi-DEER for this experiment, for ease of implementation we do not use the memory efficient version, i.e. we instantiate the full Jacobian and then take the diagonal of it. Nonetheless, we still demonstrate speed gains. Directly implementing the diagonal derivative would likely lead to further speed and memory gains.

The RNN used in this experiment is a 5 layer GRU. When we evaluate this architecture in parallel, we evaluate each layer in parallel using (quasi)-DEER. In Figure 3 (right), we report the number of (quasi)-DEER iterations averaged over all layers and batches. In Figure 7, we report the number of iterations needed for convergence in the last layer only.

## B.3 ELK and Quasi-ELK for Evaluating Autoregressive RNNs

Here we discuss our experimental details for our Experiment 3, discussed in Section 6.3.

### B.3.1 AR GRU Architecture

The architecture is a GRU with hidden states $\mathbf{h}_t \in \mathbb{R}^3$ and scalar inputs $x_t \in \mathbb{R}$. However, at every point in the sequence $t$, we readout the hidden state $h_t \in \mathbb{R}^3$ and use it to parameterize a mean $\mu_{t+1} \in \mathbb{R}$ and a variance $\sigma_{t+1}^2 \in \mathbb{R}_+$. We then sample $x_{t+1}$ according to $x_{t+1} \sim \mathcal{N}(\mu_{t+1}, \sigma_{t+1}^2)$; this *output* $x_{t+1}$ is then fed into as the *input* to the AR GRU at time step $t+1$ to make the new hidden step $\mathbf{h}_{t+1}$.

This AR GRU is trained using standard sequential evaluation and backpropagation-through-time to produce a noisy sine wave of length 10,000. We train the AR GRU on 1024 traces $\mathbf{x}_{1:T}$ generated from a sine wave with amplitude 10 and white noise applied to each time step, and the training objective is to minimize the the negative log probability of the $\mathbf{x}_{1:T}$.

Once the AR GRU has been trained, it can generate its own trace $\tilde{\mathbf{x}}_{1:T}$ given an initial hidden state $\mathbf{h}_0$ and noises $\boldsymbol{\epsilon}_{1:T}$.

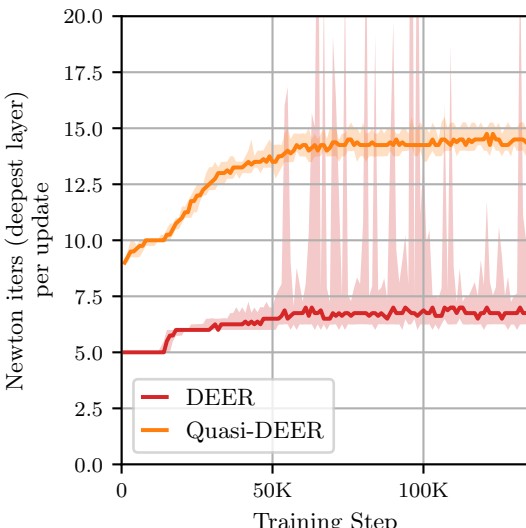

**Figure 7:** The number of Newton iterations needed for the deepest layer of the 5 layer GRU used for the time-series classification task in Section 6.2.

We note that such a system is Markovian with dimension $D = \dim(\mathbf{h}) + \dim(x)$, as together the hidden state $\mathbf{h}_t$ and output $x_{t+1}$ determine the next hidden state $\mathbf{h}_{t+1}$ and output $x_{t+2}$. Thus, in the notation of Section 2, a hidden state $\mathbf{s}_t$ of the Markovian state space model is $\mathbf{s}_t = (x_{t+1}, \mathbf{h}_t)$. Therefore, we can apply fixed-point methods to try to find the correct trace $\mathbf{s}^*$ in a parallelized manner instead of autoregressively.

We note that one distinction of this set-up with respect to the notation developed in Section 2 is that the dynamics functions $f$ are effectively *time-varying* because the way in which $x_{t+2}$ is generated from $(x_{t+1}, h_t)$ depends on the noise $\epsilon_{t+2}$, the value of which varies across the sequence. However, all the results in the paper still apply after subsuming the input dependence into a time-varying dynamics function $f_t$.

### B.3.2  Wall-clock Time Benchmark

The timing experiments were carried out as follows on an Nvidia A100 GPU with 80 GB of GPU memory, using python 3.9 and jax 0.4.11. Exact wall-clock times will vary depending on hardware, software, and implementation.

We ran sequential evaluation of the trained AR GRU to produce noisy sine waves of length $T = 10{,}000$, as well as the four parallelized method we consider in this paper (DEER, quasi-DEER, ELK, and quasi-ELK).

We ran 20 different random seeds (which lead to different values of the $\epsilon_{1:T}$ and therefore different nonlinear dynamics), and timed each for a total of 4 repetitions (i.e. 80 timing runs per method). We record the wall-clock time needed to evaluate the length $T$ sequence sequentially, as well as wall-clock time, divided by $T$ needed to run $T$ Newton iterations of each of the parallelized methods (thus, we obtain the time per Newton iteration for each of the parallelized methods).

Over these 80 timing runs, the sequential evaluation took an average of **96** milliseconds, with standard deviation of $1.55$ ms. We report the average time per Newton iteration, the total number of iterations needed for convergence, and the total wall-clock time to convergence in Table 2. Note that the third column of Table 2 is the product of the first two columns.

We effectively read the number of Newton iteration to convergence off of the graphs in Figure 4, but find the number of Newton iterations to convergence to be quite stable across random seeds (see Figure 8).

These timing results are illustrative of multiple themes of our paper. We see that while the undamped Newton steps are individually faster because they are carrying out fewer computations (they are just computing a linear recurrence relation, or equivalently an undamped Newton step, instead of

**Table 2:** Time to evaluate a length $T = 10,000$ trained AR GRU using sequential vs parallelized methods. We note the `dynamax` package [27] we used for the parallel Kalman filter implementation in ELK is not optimized for speed, and hence these run times could be further improved.

| Algorithm | Time per Newton step (ms, mean $\pm$ std) | Newton steps to convergence | Total time to convergence (ms) |
|---|---|---|---|
| **Sequential Evaluation** | | | |
| Sequential | N/A | N/A | 96 |
| **Parallelized Methods** | | | |
| DEER | $0.282 \pm 0.0005$ | 4449 | 1255 |
| Quasi-DEER | $0.087 \pm 0.0002$ | 7383 | 642 |
| ELK | $3.600 \pm 0.0670$ | 172 | 619 |
| Quasi-ELK | $0.141 \pm 0.0004$ | 1566 | 221 |

computing a filtering pass, or equivalently solving a trust region problem). However, because the undamped Newton methods are numerically unstable, they take dramatically more Newton steps to convergence.

Similarly, we see that the quasi methods are dramatically faster than their dense counterparts as they are replace $\mathcal{O}(D^3)$ matrix-matrix multiplication with $\mathcal{O}(D)$ diagonal matrix multiplication. The $\mathcal{O}(D^3)$ work required by a parallel scan on a dense linear recurrence likely saturates the GPU). We see in Table 2 that individual steps in the dense DEER/ELK are (approximately) a factor of between 3.5 and 30 times slower *per step* than their quasi (diagonal) variants. However, they take a factor of between 2 and 10 fewer iterations.

Thus, we find that our fastest parallelized method on wall-clock time is quasi-ELK, but even so it is approximately two times slower than sequentially evaluating this AR GRU. Therefore, an interesting direction for future work would be to characterize regimes where parallel methods can outperform sequential methods, and to investigate whether this autoregressive setting is such a regime, or whether parallelized methods can benefit from further speed-ups by leveraging adaptive trust region sizes, clever initialization strategies, or even more modern parallelized hardware.

### B.4 Setting the Hyperparameter for the AR GRU

We provide more details on how to set the hyperparameters for ELK. Figure 8 shows how to set the hyperparameter for ELK in the context of the evaluating the AR GRU that generates a noisy sine wave (Figure 4).

We sweep over the hyperparamter for 15 different input sequences, and plot the median and quartiles of the cost to convergence in terms of Newton iterates and runtime (left column of Figure 8). We see a bathtub curve: large $\lambda$ takes needlessly small steps, slowing progress; small $\lambda$ results in many resets, slowing convergence. Crucially, we see there is little variance across individual sequences. These results show that there is a well-behaved dependence that can be optimized on a validation set with a simple 1-d grid search.

We also chart the approximation error against cost for the AR GRU (center and right column of Figure 8). We see that the approximation error reduces in fewer Newton steps with full DEER as opposed to quasi-DEER, but, crucially, the wall-clock time (the more important of the two metrics) is notably lower across all accuracies for quasi-DEER. This indicates that our more efficient – but approximate – quasi-DEER is broadly preferable to the more expensive – but exact – DEER updates. Furthermore, the stabilized ELK and quasi-ELK are better still. We also show the steps/time to convergence for a range of accuracy thresholds, and see that our methods outperform DEER across the full range of thresholds and metrics.

These experiments were run on a single Nvidia A100 with 80GB of onboard memory.

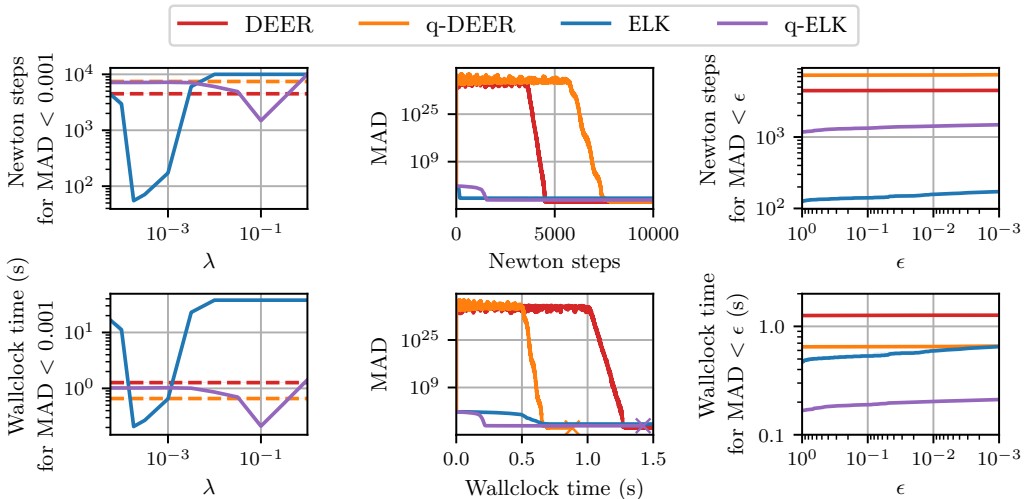

**Figure 8:** Experiment to show how to set the hyperparameters for (quasi)-ELK on the AR GRU pre-trained to generate a noisy sine wave (Figure 4 in the main text). Top row plots Newton steps; bottom row plots wall-clock time. Lower is better for all plots. (**Left**) median steps/time to convergence over $\lambda$ over 15 sequences. Quartiles are shaded but are very small. DEER methods are independent of $\lambda$. (**Center**) Updated version of Figure 4 instead plotting MAD as a function of wall-clock time. (**Right**) Time to convergence is robust as a function of convergence threshold $\epsilon$. Median and quartiles across 15 sequences are shown. DEER methods are nearly constant at the thresholds considered (very slight positive slope). Note we plot for *increasing* $\lambda$ corresponding to a smaller trust region, and *reducing* $\epsilon$ corresponding to a tighter convergence threshold.

## B.5 Additional Experiment: Evaluating Chaotic Lorenz96 Systems

We include an extra experiment where we tackle the parallel evaluation of the classic non-linear 5-dimensional Lorenz-96 system, with $F = 8$ which results in chaotic dynamics. We seek to evaluate this system (for $T = 1000$ timesteps) using (quasi)-DEER and (quasi)-ELK. We directly use the Lorenz-96 dynamics as our nonlinear dynamics function $f$, i.e. the architecture/time evolution *is* the Lorenz-96 ODE system. The state is the five-dimensional Lorenz system state. The input is therefore the initial condition of the ODE; and the outputs are the $T \times 5$ subsequent system states.

We demonstrate that all the parallelized methods converge to the correct trace, but that (quasi)-ELK is dramatically more stable at intermediate Newton iterations prior to convergence. We see that DEER and ELK methods converge in a comparable number of steps (this makes sense as DEER is a special case of ELK for $\lambda \to 0$). DEER is faster (in terms of wall-clock time) because of the extra work done per ELK iteration. However, ELK has stabilized convergence, whereas DEER relies heavily on resetting. Interestingly we see that quasi is slower by all metrics, suggesting that the chaotic dynamics may require the more accurate updates. Quasi methods can be implemented to consume notably lower memory, however, and so may be preferable in certain circumstances.

In Figure 9, we report mean absolute deviation (MAD) of the time series at Newton iteration $(i)$ against the true state sequence. "Iteration" then refers to the number of Newton iterations, i.e. the number of updates applied to the entire state sequence. We set hyperparameters using 10 different evaluations of the Lorenz96 (i.e. starting from 10 different initial points).

These experiments were run on a single Nvidia A100 with 80GB of onboard memory.

## B.6 Background on Parallel Scans

For a more detailed reference on parallel scans, the interested reader should refer to Appendix H of Smith et al. [5] or to Blelloch [24].

In our codebase, we leverage `jax.lax.associative_scan` with the correct binary associative operator. The binary associative operator for DEER and quasi-DEER is simply the composition of

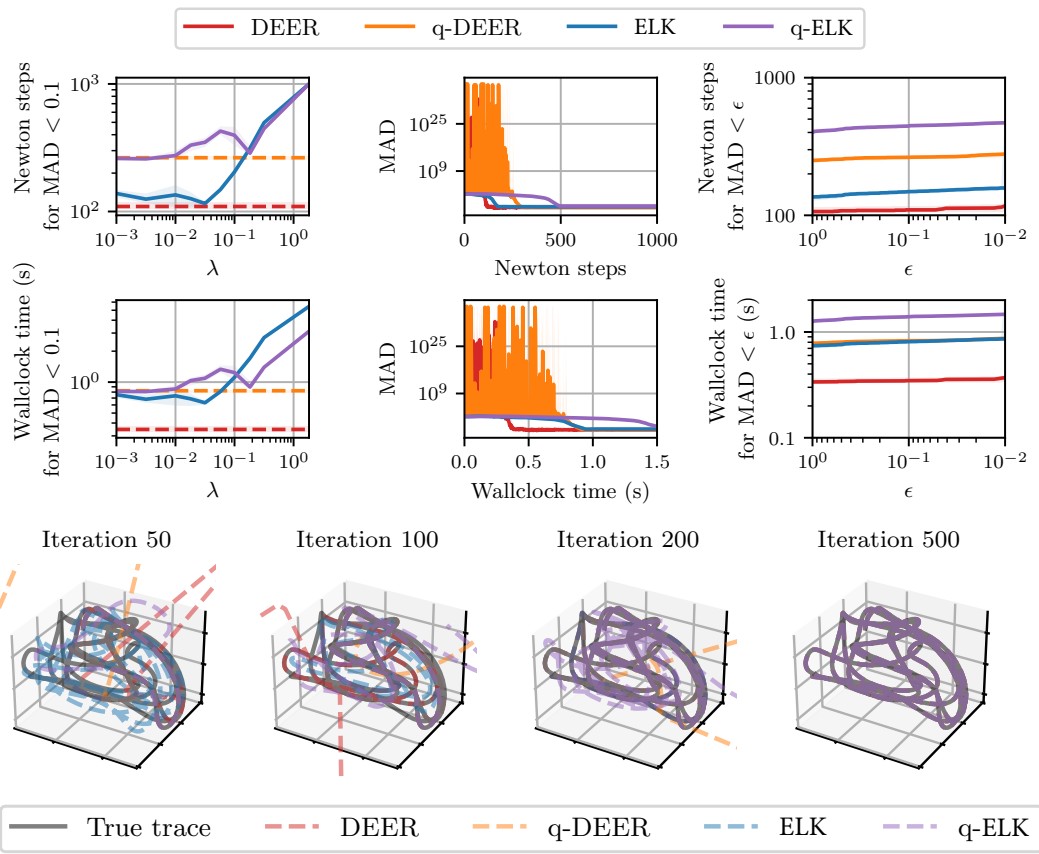

**Figure 9:** Evaluating the Lorenz96 system in parallel. **(Top two rows)**: Same format as Figure 8. **(Bottom row)**: Plot of Lorenz96 trajectory during optimization. DEER methods are noticeably more unstable than ELK methods.

affine maps, while the binary associative operation for Kalman filtering can be found in Särkkä and García-Fernández [26] and in dynamax [27].

## C   Additional Background on Newton's Method

In this appendix, we provide additional background on Newton's method, and why it is of use for parallelizing nonlinear RNNs.

Newton's method provably enjoys quadratic (very fast) convergence in a basin near the true solution. Moreover, as exhibited by the widespread usage of Newton's method across many domains, Newton's method can exhibit fast convergence in practice. However, a major motivation for this paper is that globally, Newton's method can be unstable and converge slowly. This instability is a major motivation for our development of ELK.

A core insight from Lim et al. [1] is that in the setting of evaluating RNNs, Newton's method can be cast as a parallel scan (called DEER). At each "Newton iteration," DEER linearizes the nonlinear dynamics of the RNN it is evaluating. To the extent that linear approximations are a very powerful tool across a wide variety of domains (e.g. Taylor expansions), this linear approximation can be a good approximation, leading to rapid convergence. For example, if we were dealing with linear RNNs, DEER would converge in one Newton iteration. In this paper, we are instead dealing with nonlinear RNNs, so more Newton iterations are required.

## C.1 Newton's Method for Root-Finding

We provide a brief discussion of Newton's method for root finding. A great resource for further study is Nocedal and Wright [23].

Let's say we want to find the solution $\mathbf{s}^*$ to the nonlinear equation $\mathbf{r}(\mathbf{s}) = 0$, and we have a guess $\mathbf{s}^{(i)}$ at iteration $i$. Newton's method linearizes $\mathbf{r}(\mathbf{s})$ at the guess $\mathbf{s}^{(i)}$, i.e.

$$\hat{\mathbf{r}}(\mathbf{s}) := \mathbf{r}(\mathbf{s}^{(i)}) + \frac{\partial \mathbf{r}}{\partial \mathbf{s}}(\mathbf{s}^{(i)})(\mathbf{s} - \mathbf{s}^{(i)}).$$

Thus, we get our new guess $\mathbf{s}^{(i+1)}$ as the solution to $\hat{\mathbf{r}}(\mathbf{s}) = 0$. Therefore, $\mathbf{s}^{(i+1)}$ satisfies

$$\mathbf{s}^{(i+1)} - \mathbf{s}^{(i)} = -\mathbf{J}^{-1}\mathbf{r}(\mathbf{s}^{(i)}),$$

where we define $\mathbf{J} := \dfrac{\partial \mathbf{r}}{\partial \mathbf{s}}$.

## C.2 Newton, Gauss-Newton, Root-Finding, and Optimization

In this paper, we seek to find the root of a nonlinear equation $\mathbf{r}(\mathbf{s}) = 0$. In Appendix C.1 we discuss how to use Newton's method for root finding to obtain the update

$$\mathbf{s}^{(i+1)} \leftarrow \mathbf{s}^{(i)} - \mathbf{J}^{-1}\mathbf{r}.$$

However, another approach is to consider minimizing the merit function $\mathcal{L}(\mathbf{s}) := \|\mathbf{r}(\mathbf{s})\|_2^2/2$. The root $\mathbf{s}^*$ of $\mathbf{r}$ will also minimize $\mathcal{L}(\mathbf{s})$, so the goal of root-finding to solve $\mathbf{r}(\mathbf{s}) = 0$ is the same as trying to find the minimize of $\mathcal{L}(\mathbf{s})$. However, if one applies Newton's method for optimization to try to minimize $\mathcal{L}(\mathbf{s})$ (see Boyd and Vandenberghe [77] for a great reference on Newton's method and optimization), the update obtained is actually

$$\mathbf{s}^{(i+1)} \leftarrow \mathbf{s}^{(i)} - H(\mathbf{s}^{(i)})^{-1}g(\mathbf{s}^{(i)}),$$

where $\mathbf{H}$ is the Hessian of $\mathcal{L}$ and $\mathbf{g} = \mathbf{J}^T\mathbf{r}$ is the gradient of $\mathcal{L}$ with respect to $\mathbf{s}$. The Gauss-Newton method approximates this optimization update for minimizing the merit function by making the approximation $\mathbf{H} \approx \mathbf{J}^T\mathbf{J}$, and so the Gauss-Newton update for minimizing the merit function ends up being the same as the Newton update for the finding the root of $\mathbf{r}$.

## C.3 Convergence of Newton's Method

Newton's method only converges within a suitable basin [78, §1.2.4, p. 37], but establishing best practices for initialization is an open problem. For instance, Yap provides a bound on the norm of the basin [79, Lecture IV, §10, p. 174]. However, this definition requires bounding the derivative of the objective function, which is harder than the original problem. Nesterov derives a basin for quadratic convergence around the true solution [78, Thm 1.2.5, §1.2.4, p. 39], but does not provide information on how to locate this basin *a priori*. Indeed, Nesterov defaults to taking standard gradient steps early in optimization until you assume you are in the basin, and *then* using Newton steps [78, §1.2.4, p. 39].

# D   Algorithm Block

For the reader's convenience, we provide an algorithm block depicting "the ungulates"[2] (parallelized RNN algorithms, i.e. DEER, ELK, and the quasi-variants).

---

[2]An ungulate is a large hooved mammal, and our affectionate term for DEER, ELK, and the quasi-variants

**Algorithm 1** ParallelizeRNN

1: **procedure** PARALLELIZERNN($f$, $s_0$, init_guess, tol, method, quasi)
2:     diff $\leftarrow \infty$
3:     states $\leftarrow$ init_guess
4:     **while** diff $>$ tol **do**
5:         shifted_states $\leftarrow [s_0, \text{states}[:-1]]$
6:         $fs \leftarrow f(\text{shifted\_states})$
7:         $Js \leftarrow$ GETJACOBIANS($f$, shifted_states)
8:         **if** quasi **then**
9:             $Js \leftarrow$ DIAG($Js$)
10:        $bs \leftarrow fs - Js \cdot \text{shifted\_states}$
11:        **if** method = 'deer' **then**
12:           new_states $\leftarrow$ PARALLELSCAN($Js, bs, s_0$)
13:        **else if** method = 'elk' **then**
14:           new_states $\leftarrow$ PARALLELKALMANFILTER($Js, bs, \text{states}, s_0$)
15:        diff $\leftarrow \|\text{states} - \text{new\_states}\|_\infty$
16:        states $\leftarrow$ new_states
17:     **return** states

