# OpenReview forum: "Towards Scalable and Stable Parallelization of Nonlinear RNNs"
_NeurIPS.cc/2024/Conference — NeurIPS 2024 poster_

### Official Review · Reviewer_4jAU · 2024-07-12

**Soundness:** 3
**Presentation:** 3
**Contribution:** 2
**Rating:** 6
**Confidence:** 2

**Summary:**

This paper aims to address the challenge of parallelizing the evaluation of nonlinear Recurrent Neural Networks (RNNs). Key contributions include the introduction of quasi-Newton approximations and trust region-based methods to reduce computational complexity and improve numerical stability, respectively. These enhancements enable reductions in memory usage and computational time, while maintaining the accuracy of the model's evaluations. The techniques presented are empirically validated, demonstrating their efficacy and potential application across various domains where RNNs are employed.

**Strengths:**

- The integration of quasi-Newton methods provides a fresh perspective on overcoming the challenges in parallelizing the evolution of nonlinear RNNs. The theoretical insight extends the scalability of RNNs soundly.

- By employing quasi-Newton approximations, the paper effectively reduces the computational complexity from cubic to nearly linear with respect to the sequence length. This reduction is critical for deploying RNNs in resource-constrained environments.

- The introduction of trust regions and their integration with Kalman smoothing techniques stabilizes the Newton iterations. This is a notable advancement over traditional methods, which often suffer from numerical instabilities due to undamped Newton updates.

- The methods are not only theoretically sound but are also empirically validated to demonstrate their practical effectiveness. This includes significant reductions in memory usage and computational time, crucial metrics for real-world applications.

**Weaknesses:**

- The paper's empirical validation, though robust, is limited to specific architectures and scenarios. Expanding the testing to a wider range of nonlinear RNN types and more diverse datasets could provide a more comprehensive understanding of the methods’ applicability and limitations.
- The implementation of quasi-Newton and trust-region methods might be complex and could require significant modifications to existing neural network training pipelines. This complexity could hinder widespread adoption without additional tools or simplified frameworks.

**Questions:**

- As I am not very familiar with this topic, I am wondering which equation will be speeding up using parallel computing. And how the method scales when parallelism is increasing or decreasing.
- More experimental details are required such as how the parallelism is implemented and what kind of parallel computing is applied in the experiments.
- Will the word "parallel" be proper for this type of method? Initially, I thought the parallelism was w.r.t the sequential length from the title. However, it seems that parallelism is applied for the quasi-newton method, which to me is equal to speeding up the computation in optimization with parallel computing. How it is related to parallelizing RNN in terms of the sequential length?

**Limitations:**

The authors have not adequately addressed the limitations.

---

> ### Author Rebuttal · Authors · 2024-08-06
>
> Firstly, we thank the reviewer for taking the time to review our submission and for their positive comments! We were particularly pleased by the comments on the foundation, clarity and presentation of our work.
>
> > Weaknesses
>
> We discuss these in global review (“empirical validation” in Section 2 and “implementation” in Section 1.1). Broadly, regarding “empirical validation” we appreciated your perspective to bring our robust experimental validation to other datasets, and so took the suggestion of Reviewer VSf9 and demonstrated our methods on the chaotic Lorenz-96 system. Regarding “implementation,” we note that our method can be dropped into any stateful architecture.
>
> > As I am…
>
> This is a very interesting point, that we probably didn’t discuss in enough detail in the paper. The key insight of DEER was remapping the matrix inversion (a prohibitively expensive operation) required by the Newton iteration into a parallel scan. This allows the “inverse” to be solved efficiently and while exploiting parallelism. However, the parallel scan incurs $O(TD^3)$ work, and hence does not scale well (and is prone to instabilities due to the unboundedness of the individual Jacobians). Our insight was to introduce quasi-approximations into this, to reduce this complexity to $O(TD)$, while proving this retains global convergence (c.f. Proposition 1).  (This suggested the “resetting” heuristic for combating instabilities; and also led us towards the connection with trust region stabilisation.)
>
> The parallel scan, given sufficiently many processors, scales as $O(\log T)$. As we show in Figure 6, we see this speedup at low model sizes and sequence lengths. Once the processors are saturated, we see a linear increase in the runtime (since the amount of work we do is linear), but it is making much more effective use of the GPU, resulting in a constant factor speedup over sequential application at larger model sizes/sequence lengths.
>
> We have added an expanded discussion of these points, thank you for highlighting them.
>
> > More experimental details…
>
> The JAX library includes an implementation of parallel scan which we use off-the-shelf. As a result the implementation (with regard to parallelism) is very straightforward. We included code in the original submission and are also preparing a full public code release. We have also added more tabulated hyperparameters, configs, runtimes etc; and added a paragraph explaining parallel scans in detail and how we leverage them.
>
> > Will the word…
>
> We believe “parallel” is the correct word. Crucially, we pose the inherently sequential operation as an optimization over the length of the sequence, where each optimization step is parallelizable (sub-linear parallel application time in the sequence length, $O(\log T)$, to be exact). We have reinforced this point in the introduction. If we have misunderstood the reviewers query, then please let us know!
>
> > The authors have…
>
> We refer the reviewer to the general response in regard to discussion of limitations. We have added much more explicit discussion of the limitations in a single place to correct this shortfall. If the reviewer has additional points they would like us to include, then please let us know, we are more than happy to include them!
>
> We again thank the reviewer for taking the time to review our submission and for raising some really interesting points of confusion that we had not foreseen. Fixing these has definitely made the paper better. Please let us know if you have any further questions!
>
> — Submission 13253 Authors

---

> > ### Comment · Reviewer_4jAU · 2024-08-10
> >
> > Thank you for your helpful clarification and I would like to upgrade my score. I suggest the authors add these discussions in the revised version.

---

> > > ### Author Response · Authors · 2024-08-10
> > >
> > > Thank you for your helpful comments. We agree that they have made the paper better!
> > >
> > > Thank you as well for upgrading your score and for advocating for acceptance of this paper!

---

### Official Review · Reviewer_KXRw · 2024-07-13

**Soundness:** 3
**Presentation:** 4
**Contribution:** 3
**Rating:** 6
**Confidence:** 3

**Summary:**

In this paper, the authors propose to improve DEER, a previous method that evaluates non-linear RNN in parallel by viewing it as a fixed point problem. Specifically, instead of using Newton's method to solve the fixed point problem, the authors leverage quasi-Newton methods to approximate the procedure. This alleviates the intractability of DEER. Moreover, they stabilize DEER by connecting it to Kalman smoothing technique. Empirical results show that the improved methods demonstrate superior performance in both speed and memory usage.

Strengths

1. The paper is well written and easy to follow.
2. The proposed methods are very well justified.
2. The proposed technique is reasonably designed.

Weaknesses

1. The experiments are thin.
2. Although the proposed methods are novel, the adopted techniques are already well-known.

Overall, I recommend a weak acceptance of this paper.

**Strengths:**

1. The paper is well written and easy to follow.
2. The proposed methods are very well justified.
2. The proposed technique is reasonably designed.

**Weaknesses:**

1. The experiments are thin. In this paper, the authors evaluate the quality of different methods on only one dataset with a fixed model. It is hard to predict how the proposed methods will behave on other datasets with various RNN architectures.
2. Although the proposed methods are novel, the adopted techniques are already well-known. For example, quasi-Newton methods and the Levenberg-Marquardt update are well-studied before. This weakens the novelty of this paper.

**Questions:**

See the weaknesses.

**Limitations:**

The authors have adequately addressed the limitations.

---

> ### Author Rebuttal · Authors · 2024-08-06
>
> Firstly we thank the reviewer for taking the time to review our submission and for their positive comments! We were particularly pleased by the comments on the foundation, clarity and presentation of our work.
>
> The central reservation of the reviewer is that our experimental results are “thin”. In regard to this, we refer the reviewer to the general response and the supplemental review PDF, where we have included additional experiments.
>
> The reviewer also comments that many of the components we discuss exist in the literature.  We ultimately agree with this statement; but argue that the real merit in our paper is assembling these components and exploring (theoretically and empirically) their relative strengths and weaknesses, and stratagems to operationalizing these components in the context of deep learning and non-linear systems.  As such, we believe the ties to existing work highlights the timeliness and utility of our paper – as we clearly build directly and pragmatically on existing literature that is of general interest to many in the community.
>
> We again thank the reviewer for their evaluation of our paper, their kind words, and for pointing to avenues to improve the paper. Please let us know if you have any further questions!
>
> — Submission 13253 Authors

---

> > ### Comment · Reviewer_KXRw · 2024-08-11
> >
> > Thank you for your response. I remain my stance to accept this paper.

---

> > > ### Author Response · Authors · 2024-08-13
> > >
> > > Thank you for advocating to accept this paper!

---

### Official Review · Reviewer_5uzK · 2024-07-14

**Soundness:** 4
**Presentation:** 4
**Contribution:** 3
**Rating:** 7
**Confidence:** 4

**Summary:**

This paper presents improvements over the approach of Lim et al. last year for the parallelization of nonlinear recurrences in the sequence length. The paper jumps right into the problem after a quick introduction, and presents a few improvements on Deer: Quasi-Deer (an approximate method), IKE (stable but expensive), and Quasi-IKE (moderate stability, moderate memory and work).
The authors corroborate their improvements with some theoretical results and experiments

**Strengths:**

The paper flows very well and jumps right into the math: the authors have lots to show and I must say I was impressed by the quality and flow of the paper. While I have some questions and think this line of work does not end with this paper, I believe this contribution is worth publishing. I think the author correctly shows their improvements, and I particularly liked the quasi-deer approach. I also really like how the paper is hones on limitations and clearly provides a gateway to future research.

**Weaknesses:**

I will put here weaknesses and questions at the same time:

1) Approximation quality: One missing plot is the approximation error: even in a toy setting and with random input, it is interesting to understand how well the methods can approximate the nonlinear RNN and at which cost. You have something similar in Figure 3, but this is not enough: I would be very curious to see, e.g., the evolution of the hidden state norm.

2) Speed: My experience with non-diagonal RNNs in Jax using parallel scan is that associative scans on dense recurrences are slower than linear processing - in contrast to the diagonal setting where I usually observe something like a factor 10. You do comment a bit on this, but I am curious about how much it costs, in your framework, to do a parallel scan on dense matrices. This is of course a lower bound on your compute, right? I am not a hardware person, so this counts more as a curiosity, which I think should be better developed in the paper.

3) What is this doing?: I think that, philosophically maybe, approximations sometimes work not because they are close to the object of interest, but for some other reason. This was, for instance, the case of S4 - where the ZOH discretization introduces some structure which ends up being the reason for its success. So I ask: can you present an example where you provide a precise computation of what your recurrence is doing? I think this would be super interesting, and perhaps connect your method to some other literature on SSMs.

**Questions:**

above.

**Limitations:**

Well discussed by authors

---

> ### Author Rebuttal · Authors · 2024-08-07
>
> We thank the reviewer for taking the time to review our paper, for their positive feedback, and for providing some very interesting discussion points!
>
> > Approximation quality:
>
> This is a great observation. We have added Review PDF Figure 1 probing the compute or computational  budget required to achieve different levels of accuracy in the AR GRU experiment. There are two major findings here: (a) the speed ups are fairly robust across elements *within* a dataset (i.e. once tuned the speedups are fairly low variance); and (b) quasi-methods took more steps, but were faster as each optimization step is faster. We also find that IKE methods converged faster for the AR GRU. However, we note this relationship is reversed in the additional Lorenz96 experiments (proposed by reviewer VSf9; see global review), where IKE and DEER converge in a similar number of steps, and quasi methods require notably more. This reinforces a point made in the discussion of the original submission that different methods may be faster or more stable in different settings, but that the memory savings and ultimate convergence are consistent. We have expanded discussion and guidance in the paper.
>
> > Speed:
>
> Yes, dense matrices are much slower to process, and this matches our experience (the parallel scan on a dense linear recurrence requires $O(D^3)$ work which saturates the GPU). Table 2 in the paper shows the average time per step and average number of steps required to converge.  The key take-away is that individual steps in DEER/IKE are (approximately) a factor of between 3.5 and 30 times slower _per step_ than their quasi variants (which roughly tallies with your experience). However, they take (on average in Figure 4) a factor of between 2 and 10 fewer iterations. We have added some additional discussion of this tradeoff.
>
> > What is this doing?
>
> This is a super interesting question. We have two notable observations about (re-)interpreting the methods we present.  Below we include a condensed proof on how IKE can be re-interpreted as stabilising the parallel scan by attenuating the eigenvalues used in the parallel scan, and hence stabilising their cumulative product.  We will add the full proof to the main paper.
>
> To your question on quasi: we don’t have a particularly satisfactory answer.  Very early in our investigations when exploring quasi approximations, we noticed that the off-diagonal elements were often much smaller in magnitude than the diagonal elements (which makes intuitive sense for many recurrent models). This gave us some confidence to explore quasi approximations further, but we were unable to derive any deeper insight, and so validated it empirically. It is a really interesting avenue of future work looking into this, or other families of approximations and their relative merits and drawbacks.
>
> **Summary**:  We again thank the reviewer for positive feedback, and for raising some really unique discussion points. Please let us know if you have any further questions!
>
> — Submission 13253 Authors
>
>
> —
>
>
> **Condensed additional proof that IKE attenuates the largest eigenvalues in linear recurrence**:  (We include a full proof in revised paper)
>
> Let $\{J_t\}$ be the Jacobians used in the linear recurrence relations and $\{b_t\}$ be the offsets. Then the prediction step of the Kalman filter (IKE) is the same as DEER. However, after applying the update step in IKE (which imposes the trust region), we obtain a second linear recurrence relation where the linear operator is given by $\Gamma_t J_t$.  It can be shown that $\Gamma_t$ is a symmetric positive definite matrix with eigenvalues bounded above by $\frac{1}{1 + \lambda}$. Thus, making using of the Spectral Theorem, it follows that the norms of the eigenvalues of $\Gamma_t J_t$ are bounded above by the max of the norms of the eigenvalues of $J_t$, scaled by $\frac{1}{1 + \lambda}$.  Note that larger $\lambda$ corresponds to more regularization/smaller trust region; and therefore correspondingly results in smaller effective eigenvalues in the scan.  We recover DEER exactly if $\lambda = 0$.  Thus, while large eigenvalues in $J_t$ are the cause of the instability of DEER when evaluating unstable dynamical systems, IKE directly attenuates these large eigenvalues, explaining why the intermediate iterations using IKE remain stable.

---

> ### Comment · Reviewer_5uzK · 2024-08-09
> **Thanks!**
>
> Thanks for the rebuttal. Very interesting - especially your view on what is the method doing to eigenvalues in practice. Keeping my accept score. Good luck!

---

> > ### Author Response · Authors · 2024-08-10
> >
> > Thank you so much for advocating for the acceptance of this paper!

---

### Official Review · Reviewer_VSf9 · 2024-07-15

**Soundness:** 3
**Presentation:** 3
**Contribution:** 2
**Rating:** 5
**Confidence:** 3

**Summary:**

This work extends the parallel RNN (DEER) method to improve its efficiency and stability. The authors make two main modifications:
1. Replace the full Jacobian matrix inverse with its diagonal, allowing for linear-time inversion.
2. Introduce damping via Kalman filter to enhance the stability of Newton methods.

**Strengths:**

- Solid contributions to improving the DEER method
- Potential for increased efficiency and stability in parallel RNN implementations

**Weaknesses:**

Overall, the paper have solid contributions. But there are several weakness

1. The potential drawbacks of these modifications are not fully explored
   - Efficiency gains may come at the cost of slower convergence
   - Damping could potentially compromise accuracy

2. Limited experimental scope
   - The experiments may not reveal issues that could arise in harder, larger-scale cases
   - For example: does the model work on chaotic systems such as Lorenz 96?

3. it will be interesting to compare with popular models like transformers and xLSTM

**Questions:**

Harder examples: how does the model work on chaotic system such as Lorenz 96 or Weather forecast?
More benchmark, how does the model compared to transformers and xLSTM?

**Limitations:**

The limitations are not sufficiently addressed.

---

> ### Author Rebuttal · Authors · 2024-08-07
>
> We thank the reviewer for taking the time to review our paper and providing some helpful feedback!
>
> > Limited experimental scope
>
> We point the reviewer to the global response for additional experiments we have added:
>
> - In Review PDF Figure 1 we study the sensitivity to the $\lambda$ parameter (related to Reviewer S8Ch’s feedback), showing that $\lambda$ can be effectively set with a 1-d hyperparameter sweep, and that actualized speed-ups for a selected $\lambda$ are robust for a given data modality.
>
> - We also add additional experiments exploring the wallclock time/iteration/accuracy tradeoff of the methods (r.e. Your comment “The potential drawbacks…”). In Review PDF Figure 1, we show for the AR GRU that while full methods take fewer steps, quasi methods converge faster in wallclock time across a broad range of desired accuracy levels.
>
> - We really liked your suggestion of using our algorithms to perform inference in a Lorenz96 system. We include results for this experiment in the Review PDF Figure 2. We see that DEER and IKE methods converge in a comparable number of steps (this makes sense as DEER is a special case of IKE for $\lambda \to 0$). DEER is faster (in terms of wallclock time) because of the extra work done per IKE iteration.  However, we do see that IKE has stabilised convergence, whereas DEER relies heavily on the resetting heuristic.  Interestingly we see that quasi is slower by all metrics, suggesting that the chaotic dynamics may require the more accurate updates.  Quasi methods still consume notably lower memory, however, and so may be preferable in certain circumstances.
>
> > It will be interesting to…
>
> The comparison to alternative architectures, such as the xLSTM and Transformers, is a potential point of comparison. However, we stress that this paper is about accelerating and operationalizing the parallel evaluation of sequential, non-linear models. Papers developing or benchmarking natively parallelizable architectures (e.g. Transformers) often state that sequential architectures such as (e.g.) GRUs cannot be parallelized, and are often therefore eliminated from consideration or score poorly on speed tests. However, DEER raised the point that this is not entirely accurate, and that alternative black-box methods may be within reach for efficiently evaluating these oft-maligned architectures. Our work focuses on extending and operationalizing DEER, without changing the architecture or its asymptotic performance. We have clarified this in the main text.
>
> > The limitations are…
>
> As mentioned in the global review, we have further highlighted and discussed the limitations in separate paragraphs in the conclusion (while also retaining the “inline” discussion to retain flow). If the reviewer has further limitations we have not addressed, then please let us know, we are more than happy to include them!
>
> We again thank the reviewer for evaluation, and for raising many points that have improved the paper. Please let us know if you have any further questions!
>
> — Submission 13253 Authors

---

> > ### Comment · Reviewer_VSf9 · 2024-08-10
> >
> > Thank you for including the Lorenz experiment. However, I’m a bit confused about how the task is defined. Typically, the task involves predicting the next state auto-regressively given the history and the iteration is the time stepping. Since the Lorenz system is ergodic, the state should not converge to a stationary state. However it seems the iteration is something different here. Could the authors provide more details about the experiment?
> > - Specifically, what are the input and output of the model?
> > - if the model is not auto-regressive, how does it modeling the time evolution?
> > - what error metric is being used?

---

> > > ### Author Response · Authors · 2024-08-11
> > >
> > > Thank you for asking follow-up questions. We were very tight on characters so we couldn’t include much information initially! We can certainly add some more clarity:
> > >
> > >
> > > > If the model is not auto-regressive, how does it model the time evolution?
> > >
> > > We tackle the parallel evaluation of the classic non-linear 5-dimensional Lorenz-96 system, with F=8 which results in chaotic dynamics.  The setup is the same for (q-)DEER and (q-)IKE. We directly use the Lorenz-96 dynamics as our nonlinear dynamics function $f$, i.e. the architecture/time evolution _is_ the Lorenz-96 ODE system.  The state is the five-dimensional Lorenz system state (instead of using an RNN approximator with its own hidden state predicting the next timestep of the true system).
> > > We chose this in part to explore the parallel evaluation of other architectures; but also to allow us to focus on the parallel evaluation of the Lorenz-96 system itself – as opposed to the capability of a function approximator to approximate the system.
> > >
> > > > Specifically, what are the input and output of the model?
> > >
> > > The input is therefore the initial condition of the ODE; and the outputs are the $T \times 5$ subsequent system states (plotted in Review PDF Figure 2, bottom row).
> > >
> > > > What error metric is being used?
> > >
> > > Our solvers [(q)-DEER and (q)-IKE] are optimizing the merit function defined in eq. (7), which is the sum of squares of the one-step prediction error (defined in eq. (1)). However, this loss expresses the pairwise consistency of consecutive inferred states under the specified dynamics;  as opposed to the more conventional notion of “accuracy”: the difference between the inferred sequence and the true sequence. We therefore plot this as the mean absolute deviation (MAD) of the time series at Newton iteration $(i)$ against the true state sequence. “Iteration” then refers to the number of Newton iterations, i.e. the number of updates applied to the entire state sequence.
> > >
> > > We hope this clears any points of confusion. Please let us know if you have any further questions!
> > >
> > > -- Submission 13253 Authors.

---

> > > > ### Comment · Reviewer_VSf9 · 2024-08-11
> > > >
> > > > Thanks for the clarification. Two more questions:
> > > > - what is T in this case? Can these algorithms work for large T such as 1000 or 10000 (which are common in chaotic system).
> > > > - Is it correct that the DEER algorithm, while unstable at the first 100 iterations, converges the fastest in wall time? Does it imply DEER is more efficient compared to the proposed methods?

---

> > > > > ### Author Response · Authors · 2024-08-13
> > > > >
> > > > > > What is T?
> > > > >
> > > > >
> > > > > Yes! $T=1000$
> > > > >
> > > > > > Is it correct?
> > > > >
> > > > > Yes, on the Lorenz task, DEER converges faster on wallclock time. However, DEER is a special case of IKE with $\lambda=0$, and the rebuttal pdf shows how to set this hyperparameter robustly.
> > > > >
> > > > > We discuss this point in more detail in our original rebuttal to you, in the bullet point about the Lorenz experiment, beginning with “We really liked your suggestion of using our algorithms to perform inference in a Lorenz96 system.”
> > > > >
> > > > > Please let us know if we can provide additional clarification!

---

### Official Review · Reviewer_jU7g · 2024-07-16

**Soundness:** 2
**Presentation:** 3
**Contribution:** 2
**Rating:** 3
**Confidence:** 4

**Summary:**

This paper presents a stable and scalable method to parallelize evaluation of an RNN across the sequence length. Furthermore, the paper presents a sketch of when the more recently proposed DEER algorithm converges. Alongside these contributions, the paper presents experiments showing the efficacy of the proposed method.

**Strengths:**

+ Identifies a shortcoming with the DEER algorithm
+ Presents a computationally efficient and scalable alternative to DEER

**Weaknesses:**

- The result proving the convergence of DEER isn't quite rigorous/general -- it relies on the Jacobians being finite which is quite a strong assumption which isn't true in practice, particularly in situations when the eigenvalues are > 1. The paper sweeps this under the rug through re-initialization, which suggests it isn't quite a general statement.

**Questions:**

- In general, getting to exactly zero error is impractical (floating point errors among other reasons) so how can the theorem be generalized to considering number of iterations to achieving \epsilon error? As a side consequence, can the authors clarify when DEER can achieve quadratic convergence rate of Newton method? How sensitive DEER is to blowing up errors when propagating information across sequence length?

**Limitations:**

Yes.

---

> ### Author Rebuttal · Authors · 2024-08-07
>
> We thank the reviewer for highlighting some interesting clarification points.
>
> > The result proving…
>
> This is a really interesting line of thought that we will clarify in the paper.  There are lots of points that your comment touches on, so we will try and answer them sequentially:
>
> **Re: Generality of Prop 1 with regard to eigenvalues**: We state in Prop. 1 that we assume the Jacobians must be *finite.* Given this assumption, the proof by induction in Appendix A.1 is rigorous and general for real numbers. Thanks to your feedback, we will include an explicit pointer to the full proof.
>
> The finiteness of the Jacobians is a core requirement for DEER. Thus, Prop. 1 (a statement about the DEER algorithm) is applicable anywhere that DEER is applicable. We also argue that the finiteness of the Jacobians is a weak assumption, as it is a requirement for training models with backpropagation and is satisfied by many commonly used architectures.
>
> However, it is absolutely true that the eigenvalues can be greater than one, and this leads to the *product* of the Jacobians computed by the scan blowing up in magnitude. In fact, the sensitivity of DEER to blowing up is directly linked to the eigenvalues of the Jacobians. It is this explosion that causes overflow on finite-precision architectures.
>
> The resetting heuristic combats this, and is motivated by a result derived as part of Proposition 1: that the first $t$ steps have zero residual after $t$ Newton steps (see ~Line 111 or 447), e.g., that we can restart the optimization with the correct $t$-length initialization. We can view this heuristic as allowing the Newton steps to resume when overflow is encountered, as opposed to failing or falling back to sequential evaluation.
>
> **IKE directly combats eigenvalues greater than one**:  A related point is that IKE was motivated to prevent overflow by limiting the step size in the latent state using a Kalman filter (KF). We have since made a connection from the Kalman filter back to the original scan:  the trust region/KF can be interpreted as attenuating the eigenvalues by a factor of at least $\frac{1}{\lambda+1}$, stabilising the scan _without_ the use of the heuristic (see **Proof that IKE attenuates the largest eigenvalues in linear recurrence** in response to 5uzK).  IKE combats exploding cumulative products without the use of the heuristic.
>
> **Re: discussion of the re-initialization heuristic**: we respectfully disagree that we “sweep it under the rug”. We discuss the heuristic in several places, and explicitly highlight the heuristic in Figure 4. The heuristic is a necessary intervention — not originally discussed by Lim et al. — to combat instabilities of undamped Newton in floating-point representations. We see the heuristic as a (albeit small) novel contribution, and a practical corollary of Prop. 1. We have expanded discussion of the limitations and opportunities posed by the reinitialization heuristic.
>
> > In general…
>
> **Re: connections back to floating point**: Our analysis was a theoretical exploration to develop methods we would eventually test empirically, and so we didn’t make explicit theoretical connections to floating point representations. Floating point precision can introduce instabilities into Newton through overflow and limited numerical precision in functions with large gradients. The finite precision also sets a floor on the fidelity of the convergence. These limitations are further motivation for stabilisation techniques, such as IKE, to combat the instabilities inherent to Newton’s method when executed in finite-precision.
>
> **Re: Convergence rates of Newton’s Method**: We spent a lot of time thinking about this. Newton’s method only converges within a suitable basin [Nesterov, 2018, Lectures on Convex Optimization, §1.2.4, p37] — but establishing best practices for initialization is an open problem. For instance, Yap provides a bound on the norm of the basin [Yap, Fundamental Problems in Algorithmic Algebra, 1993, Lecture IV, §10, p. 174]. However, this definition requires bounding the derivative of the objective function, which is harder than the original problem. Nesterov derives a basin for quadratic convergence around the true solution [Theorem 1.2.5; Nesterov, 2018, Lectures on Convex Optimization, §1.2.4, p39], but does not provide information on how to locate this basin apriori. Indeed, Nesterov defaults to taking standard gradient steps early in optimization until you assume you are in the basin, and _then_ using Newton steps [Nesterov, 2018, Lectures on Convex Optimization, §1.2.4, p39].  We experimented with this initially, but found it to be worse (by any metric we conceived).
>
> Quasi-methods do not, in general, enjoy quadratic convergence (but showing the global convergence of quasi-DEER was a major motivation behind Proposition 1), but are often observed to converge in practice. We have not been able to prove results about the convergence rates of the quasi methods we propose. However, our experiments show that their rates are acceptable in practice. Thus, studying quasi-DEER from a theoretical perspective is an interesting open problem. As a result of your points, we have included in the conclusion a list of open theoretical problems inspired by your questions. We have also added a brief survey on the convergence properties of Newton variants.
>
> > Limitations: Yes
>
> Is the reviewer saying there are limitations that we do not discuss? Or are that we do satisfactorily discuss limitations? If the former, please see the Global Response for information on how we have updated the paper.
>
> Thank you again for providing some really interesting and unique insights. The extra discussion has definitely improved the paper. Please let us know if you have any further questions!

---

### Official Review · Reviewer_S8Ch · 2024-07-28

**Soundness:** 3
**Presentation:** 3
**Contribution:** 3
**Rating:** 6
**Confidence:** 3

**Summary:**

This paper addresses the problem of parallel computation in RNNs, to be able to tap into the full potential highly efficient parallel machines (GPUs) that are available today: A naive inference of a recurrent model would apply each layer on the current hidden state, thereby requiring the length of the sequence $T$ sequential computations. The approach taken by this paper, which is introduced in the framework of Newton's method of root finding, proposes finding the solution to a chain of equations that, if satisfied, will verify that the found activations are equal to the sequential inference one.

Given sequence states $s_1,\dots , s_T$, the general approach (based on earlier method DEER() is to define a sequence of $T$ residuals $$r = [s_1-f(s_0),\dots, s_T-f(s_{T-1})]$$ which capture the discrepancies between successive activations and the update layer corresponding to that layer. The update rule is based on the gradient of these residuals wrt to the layer activations $$\Delta s_t^{(i+1)} = \left[\frac{df}{ds_{t-1}} (s_{t-1}^{(i)}) \right] \Delta s_{t-1}^{(i+1)} - r_t(s^{(i)})$$. Since at each step we can compute activations in parallel, we can execute this update for all $t=1,\dots, T$ in parallel. The proposition 1 states that at most $T$ updates will be needed to arrive at the correct solution. This is already introduced in DEER, but this paper goes on to address two main problems with DEER.

The linear recurrent update suffers from scalability due to the $D\times D$ size of the full Jacobians that impact both memory and time complexity, and numerical instability dueo to the singular values larger than one in Jacobian which can lead to numerical instabilities.

I had a difficult time delineating the contributions, but I believe the following to be the main contributions:

1. Quasi-DEER: using a simplified form for the Jacobian where only diagonal terms of the Jacobian are considered, leading to a linear memory and time complexity per each layer. However, this model might still suffer from the numerical instabilities.
2.  Iterative Kalman Evaluation (IKE), where the updates to the activations are "damped" by adding an L2 regularization term to the update rules, which can effectively prevent instabilities.

**Strengths:**

- The paper consideres an important problem, ie, parlallel computation of RNNs, which is of high practical importance.
- The key contributions of the paper in quaski-DEER and (quasi-)IKE are novel and address issues of scalability and numerical instability.
- To the best of my knowledge, the proposed contributions are novel.

**Weaknesses:**

I prefer to package most of my perceived weaknesses in the questions section, as they are mostly minor. Here are my main concerns:

* As authors list lambda's for IKE as a hyper parameter, it warrants a more comprehensive discussion on its choice and impact on the numerical results. Namely, an abalation study that shows how sensitive the choice of lambda is. Also, whether it is task and data-dependent it could be, or it is fairly robust across these.
* Contributions:  The manuscript would benefit from a more clear distinction on their contributions. Since the paper draws heavily from previous work that introduced DEER, it was a bit challenging for me at the end to delineate what in the paper was novel and the current paper’s contribution. While IKE and quasi-IKE and the propositions were clearly novel, other points were not as clear to me.

**Questions:**

* The proposition 1 seems to suggest that the convergence stops after T steps. but that is not any speedup over the trivial sequential computation of T steps of the RNN, so it is of little value it seems. Or am I missing something? Is this T-step iterative method still better than a simple sequential RNN computation? perhaps that’s just theory, If the practical results are better, they better be stated here?
* From seeing the figure 2, it seems like the Deer & quasi-DEER are faster, in light of my previous questions, can authors elaborate why is that the case? Is this because the actual number of steps required to stop is much less than T, or is it some hardware-related efficiency of the (quasi-)DEER implementation, vs the naive implementation?
* in my initial reading understanding, I confused the Newton's method for root finding with Newton's method for optimisation, which would involve Hessian of loss for finding the minima of a function, while the formula involving Jacobian is for finding the “root” of a function. I'm not sure if other readers would have a similar confusion, but perhaps a small note would help.
* I was missing a bit the high level intuition behind the residual sequence and how is it leading to the same sequential solution. For example, I wasn't sure if/how the equations would lead to a unique solution all the time or not, and why is that so. I think I would have understood it better
* To me it seems that the NN layer updates must be deterministic for this approach to work. Is this impression correct?
    * Also, perhaps it is beneficial for readers if authors elaborate on what types of sequential RNN architectures would be compatible with their approach for parallel computation?
* “Provided all of {df/dst−1} T t=2 are finite, then the Jacobian defined in eq. (3) is invertible and all of the eigenvalues are equal to one.” I don't understand why this is the case? Why are eigenvalues equal to one? Matrix J has off-diagonals, so it’s not quite clear why J inverse is so trivial to compute?
* I find the following poorly stated/supported as of my reading so far:
    * 137  As a consequence of our Proposition 1, replacing the Jacobians {df/dst−1}Tt=2 with an arbitrary  matrix will still result in global convergence of the resulting DEER-like algorithm in at most T  iterations.
    * Replacing J with any **arbitrary matrix**? really? Why?
    * For a similar reason: 		A straightforward way to reduce the computational cost is to replace {df/dst−1}Tt=2   with {diag (df/dst−1 )}Tt=2 ; that is, to take the diagonal entries of the Jacobians of the dynamics ,  functions.
    * Why does this work? Is this a theoretical result, or an empirically valid approach?
* Question on Figure 6: is the later linear growth of the wall clock because of the limited parallel compute on the GPU?
* section 6.1: why is convergence of a random AR stable, while convergence of a trained AR model is more unstable? Can you elaborate? Is this setup leading to the large top eigenvalue problem mentioned earlier?
* Figure 4: I’m not sure I understand top-right for some traces,
    * why the DEER (red) traces goes down at such a late iteration? I thought it was unstable and would never converge.
    * Also, the quasi-DEER (orange), goes up and then down, what does that mean? \
    * Why is the IKE going down faster first, but then quasi-IKE converges faster?

**Limitations:**

I am a little concerned about the way that limitations are currently addressed.
- In responses to the questions, authors have listed section 6.3 as a limitation, but I do not see where this section is exposing their method limitations.
- As also mentioned in questions, I would appreciate if authors make some comments on architectural limitations of this work.

---

> ### Author Rebuttal · Authors · 2024-08-07
>
> Thank you for your thorough and positive review!
>
> We refer the reviewer to the global response for discussion of an ablation study on $\lambda$ (r.e. “As authors list”) and limitations (r.e. “In responses to” and “As also mentioned”). We now respond to your other comments:
>
> > Contributions: The manuscript…
>
> We will explicitly list our contributions in the introduction.  Specifically we:
> - Show DEER is globally convergent
> - Introduce quasi-approximations to improve efficiency
> - Introduce trust regions (IKE) to stabilise DEER
> - Empirically validate the methods, and provide guidance on how to select and tune them
> - Have since explicitly shown there is a unique global solution to DEER (see “I was missing…” response) and provide a reinterpretation of how trust regions stabilise the original scan (see 5uzK response).
>
> > The proposition 1… **and** From seeing the…
>
> You are absolutely correct. Proposition 1 tackles the theoretical global convergence of the algorithm, showing we can expect convergence. The upper bound is impractical as it requires $T$ iterations; but this is a _worst case_ convergence and we often require much fewer than $T$ iterations (e.g. Figure 4a). Each Newton step requires more work than a sequential step; but fewer steps are required and the work is parallelizable. Proposition 1 was also important in developing the resetting heuristic for (q-)DEER. We have added clarification on this.
>
> > In my initial…
>
> Thank you for raising this. While DEER was inspired by the root finding perspective, IKE was inspired by the optimization perspective (see Section 4.2). We have further clarified the relationship between perspectives and their implications.
>
> > I was missing…
>
> We add a proof of the uniqueness of the solution, and global convergence to this solution, as we did not make this connection clear enough. To summarise:
>
> For a deterministic forward function and fixed inputs, there is a fixed sequence of latents and outputs. This is therefore the **only** sequence with zero residual (i.e. there is a unique sequence $s_{1:T}^*$ generated by the deterministic dynamics). Furthermore, DEER cannot get stuck at any point that is not this sequence. We prove this in Proposition 1. Another way to see this however is that each update step (eq. (5)) is equal to $J^{-1} r$. But, we have established (see “Provided all of” below) that $J$ is always invertible and so has trivial nullspace. Furthermore, the residual $r$ can only be zero at the unique solution $s_{1:T}^*$. Thus $J^{-1} r$ is nonzero everywhere except at the true solution, where it is zero. Thus, DEER cannot get stuck en route to finding the true and unique solution.
>
> > To me it seems…
>
> This is an interesting point: DEER and IKE  *can* handle stochastic models – because stochastic models become deterministic when the stochasticity is fixed i.e. has been reparameterized as in Figure 4. In fact, DEER and IKE have the same requirements as backpropagation, and so can be applied to any stateful recurrent model that admits backpropagation – a huge and practical class of models.
>
> > Provided all of…
>
> The eigenvalues of a lower-triangular matrix are equal to its main diagonal. The Jacobian in eq. (3) is the derivative of eq. (1). Each term in the vector is of the form $s_t - f(s_{t-1})$. Therefore, the Jacobian with respect to $s_t$ is the identity matrix; hence the leading diagonal is the identity; and all its eigenvalues equal one and the Jacobian is invertible (no zero eigenvalues).
>
> Efficiently solving this inverse is not tractable (see Line 92). However, we avoid computing the inverse by exploiting the structure in $J$ to instead pose eq. (5) as the recursion in eq. (6). This recursion is linear and can be solved in parallel with a scan.
>
> > I find the…
>
> We will make this discussion more clear in the paper: The elements in the sub-block-diagonal of $J$ can be replaced with arbitrary values – but the main block diagonal must remain as the identity and all other entries must be zero. Retaining convergence under modifications to the sub-block-diagonal portion is a corollary of Proposition 1, and can be seen from eq. (6): If all the states up to and including position $t-1$ at the $(i)$th Newton iteration are correct, then the update in eq. (6) at Newton iteration $(i+1)$ for position $t$ will use $\Delta s_{t-1}^{(i+1)} = 0$ (no update is required at position $t-1$), and so the update to $s_{t}^{(i+1)}$ no longer depends on the Jacobian. We will explicitly state this, as it motivates q-DEER and is, as you point out, a surprising result.
>
> We exploit this to develop q-DEER, retaining only the diagonal of the Jacobians. This reduces the parallel scan from $O(D^3)$ to $O(D)$ making each iteration faster (while still admitting global convergence as above), but needs more Newton iterations to converge due to approximate updates. We find that this trade-off often yields a faster wallclock time (see Review PDF Figure 1).
>
> Explicitly, the global convergence of q-DEER is a theoretical result (a corollary of Proposition 1), but the fast runtime of q-DEER in practice is an empirical result.
>
> > Question on Figure 6…
>
> Yes, that's a great point! We will more heavily emphasise this.
>
> > Section 6.1: why…
>
> Yes! The trained model does have larger eigenvalues, both at the true latent sequence, and at the sequences visited during the Newton iteration.
>
> > Figure 4: I’m…
>
> We use “unstable” to mirror Newton’s method parlance. Proposition 1 shows that $t \leq (i)$ have zero error, and so the instability applies only to $t>(i)$. Unstabilized methods can be arbitrarily bad in this regime until global convergence, but *may* converge before then. (q-)IKE removes the instabilities, leading to faster overall convergence. We have added clarification on this around Figure 4.
>
> Thank you for your thorough analysis, and for raising many points that have improved the paper. Please let us know if you have any further questions!

---

> ### Comment · Reviewer_S8Ch · 2024-08-10
>
> I would like to thank the authors for clarifying the questions I asked, and addressing my concerns. I believe adding several of the responses would to be added to the main manuscript and make it more readable.
>
> > The upper bound is ... Each Newton step requires more work than a sequential step; but fewer steps are required and the work is parallelizable.
>
> I can see that the theory is worst-case and in practice "fewer steps but more work" makes some sense on a very high level. But  I was hoping the authors can provide some intuition as to why there are fewer steps than $T$ needed in practice?
>
> Given that I had already a high score, I would keep my overall score as is but increase contribution score (from 2 to 3)

---

> > ### Author Response · Authors · 2024-08-10
> >
> > Thank you for reading our rebuttal, for your high score, and for advocating for acceptance of this paper. We are glad that our clarifications were helpful. We agree that your commentary has made the paper better!
> >
> > > The upper bound is...
> >
> > The intuition for why far fewer steps than $T$ are often needed in practice comes from the fact that DEER is equivalent to Newton's method. As we discuss in the rebuttal to reviewer jU7g, Newton's method provably enjoys quadratic (very fast) convergence in a basin near the true solution. Moreover, as exhibited by the widespread usage of Newton's method across many domains, Newton's method can exhibit fast convergence in practice. However, a major motivation for this paper is that globally, Newton's method can be unstable and converge slowly. This instability is a major motivation for our development of IKE.
> >
> > Another intuition is available from the scan (dynamical system) perspective. At each "Newton iteration," DEER linearizes the nonlinear dynamics of the RNN it is evaluating. To the extent that linear approximations are a very powerful tool across a wide variety of domains (think Taylor expansions), this linear approximation can be a decent approximation, and so lead to rapid convergence. For example, if we were dealing with linear RNNs, DEER would converge in one Newton iteration. In this paper, we are instead dealing with nonlinear RNNs, so more Newton iterations are required.
> >
> > Please let us know if there is anything else we can clarify!

---

> > > ### Comment · Reviewer_S8Ch · 2024-08-10
> > >
> > > Thank you. This does indeed give some intuition. I'm not sure if this intuitions were already in the paper. If not, adding them it would help the readers who aren't intimately familiar with Newton's method, to understand where is the main source of speedup.

---

> ### Author Response · Authors · 2024-08-10
>
> This is very good to hear. Yes, we will certainly flesh out these intuitions in any camera ready version, and agree that they will make the paper even better.

---

### Author Rebuttal · Authors · 2024-08-07

Firstly, we thank all six reviewers for their positive feedback and insightful comments.

We present methods for parallelizing the evaluation of non-linear RNNs, building on a recent method, DEER, from Lim et al. We first proved the global convergence of DEER. We then ameliorate DEER’s two major weaknesses: cubic computational complexity and instability. We achieve this by introducing quasi-approximations and trust regions (IKE), respectively. We evaluate all methods and find that quasi and IKE variants match or outperform DEER across a range of metrics, including accuracy, wall clock time, memory and iterations.

We first comment on the positives highlighted by the reviewers:
- **Well Received**: We feel all reviewers understood and commended the intention and merits of our submission.
- **Foundations and Connections**: Many reviewers noted how well-founded our approach is and how it dovetails nicely with recent work.
- **Quality of Presentation**: We were especially pleased by the consistent praise for the quality of our submissions presentation.

There were some critical themes shared in some reviews. We comment on these here:

1.  **Discussion of Limitations**: Despite several comments praising our discussion of limitations, some reviewers commented we did not adequately address the limitations of the methods we present. In a bid to create better flow we attempted to “inline” the limitations, addressing them as they arose. However, we clearly missed the mark here. We have added an additional paragraph to the discussion explicitly repeating those limitations (we do note that we did discuss these limitations in the original submission, but appreciate the reviewers emphasis on highlighting limitations):

    a. Quasi methods lose the quadratic convergence properties of Newton (but we show empirically that convergence is *faster in terms of wallclock time* as a result of the efficient approximation).

    b. Although motivated by Proposition 1, the quasi approximation can be significant.

    c. IKE stabilises evaluation, but, like DEER, has cubic complexity in the state dimension (quasi-IKE then combats this).

    d. The heuristic of resetting to zeros when unstable is motivated by Proposition 1, but does slow convergence in (q-)DEER methods.

    e. (q-)IKE adds an additional hyperparameter ($\lambda$).

    f. Extended discussion on when each technique is likely to yield the best results.

    g. Architectural/implementation limitations (see more below).

1.1 **On Architectural/implementation limitations**: (q-)DEER and (q-)IKE can be applied to _any_ stateful architecture with state in $\mathbb{R}^D$ with _minimal-to-no_ modifications to the overall pipeline. All methods effectively define a different `forward` method for the model with identical interfaces. Sequential/(q-)DEER/(q-)IKE can be switched between simply by changing a flag in the model class. (q-)DEER and (q-)IKE are then implemented in a model-agnostic manner.

The main architectural/implementation limitation, at least at the time of writing, is that Torch does not have an implementation of parallel scan. This is a major reason we wrote our code in JAX. However, with the explosion of parallel-scan-powered methods, it is our understanding that the Torch team is actively developing a parallel scan implementation.

2. **Experiments**: Several reviewers commented on the experimental evaluation – with some excellent suggestions of ways to strengthen them. Off the back of these suggestions, we have included some further experimental evaluation:

- **Review PDF Figure 1**: In response to the comments of S8CH, VSf9 and 5uzK, we explore $\lambda$ in (quasi)-IKE for the AR-GRU (Left Column). We sweep over $\lambda$ for 15 different input sequences, and plot the median and quartiles of the cost to convergence in terms of Newton iterates and runtime. We see a bathtub curve (large $\lambda$ takes needlessly small steps, slowing progress; small $\lambda$ results in many resets, slowing convergence).  Crucially, we see there is little variance across individual sequences. These results show that there is a well-behaved $\lambda$ dependence that can be optimised on a validation set with a simple 1-d grid search.

Furthermore, In response to the comments of S8CH, VSf9 and 5uzK, we chart the approximation error vs cost for the AR GRU (Center and Right Column). We see that the approximation error reduces in fewer Newton steps with full DEER as opposed to quasi-DEER, but, crucially, the wallclock time (the more important of the two metrics!) is notably lower across all accuracies for quasi-DEER. This indicates that our more efficient – but approximate – quasi-DEER is broadly preferable to the more expensive – but exact – DEER updates. Furthermore, the stabilised IKE and quasi-IKE are better still. We also show the steps/time to convergence for a range of accuracy thresholds, and see that our methods outperform DEER across the full range of thresholds and metrics.

- **Review PDF Figure 2**: In response to the comments from VSf9, KXRw and 4jAU, we include an extra experiment applying (q-)DEER and (q-)IKE to approximate the chaotic Lorenz-96 system. We demonstrate that all the parallelized methods converge to the correct trace, but that (q)-IKE are dramatically more stable at intermediate Newton iterations prior to convergence.

> Explicit pointer to full proof of Prop1 in Appendix A.1

We noticed that we forgot to include a pointer in the main text to the full proof of Prop1, which can be found in Appendix A.1 of the original submission. We now include this explicit pointer.

**Summary**: We hope that the expanded and clarified discussion of limitations and these additional experiments allay the reviewers' concerns, and help further elucidate the relative benefits of the methods we introduce. Thank you again, and please do not hesitate to reply if there are further clarifications we can make!

— Submission 13253 Authors

---

### Decision · Program_Chairs · 2024-09-25

**Decision:**

Accept (poster)

**Comment:**

Parallelized RNNs require solving a fixed-point equation. A previous study proposed solving this equation using Newton's method. This paper improves the computational cost and stability of Newton's method for the parallelization of RNNs. To reduce computational cost, the paper proposes a quasi-Newton method with significantly lower per-iteration complexity. To enhance stability, a trust region-based method combined with Kalman smoothing is utilized. According to reviews, the paper is well-written, and the contribution is sound. To address concerns about the experiments, the authors provided additional experimental results that elaborate on the stability enhancements. While there are concerns about comparing the proposed method with DEER, it is difficult to establish a good convergence rate for quasi-Newton methods, which makes the convergence-cost tradeoff challenging. According to reviewers, the contributions are sufficient for acceptance.